# A Probabilistic Framework for Modular Continual Learning

**Lazar Valkov[1],[*] Akash Srivastava[1], Swarat Chaudhuri[2], Charles Sutton[3]**
[1]MIT-IBM Watson AI Lab, [2]UT Austin, [3]University of Edinburgh

## Abstract

Modular approaches that use a different composition of modules for each problem are a promising direction in continual learning (CL). However, searching through the large, discrete space of module compositions is challenging, especially because evaluating a composition's performance requires a round of neural network training. We address this challenge through a modular CL framework, PICLE, that uses a probabilistic model to cheaply compute the fitness of each composition, allowing PICLE to achieve both perceptual, few-shot and latent transfer. The model combines prior knowledge about good module compositions with dataset-specific information. We evaluate PICLE using two benchmark suites designed to assess different desiderata of CL techniques. Comparing to a wide range of approaches, we show that PICLE is the first modular CL algorithm to achieve perceptual, few-shot and latent transfer while scaling well to large search spaces, outperforming previous state-of-the-art modular CL approaches on long problem sequences.

## 1 Introduction

*Continual learning* (CL) (Thrun & Mitchell, 1995) demands algorithms that can solve a sequence of learning problems while performing better on every successive problem. A CL algorithm should avoid catastrophic forgetting, i.e., not allow later problems to overwrite knowledge learned from earlier ones, and achieve transfer across a large sequence of problems. It should also be plastic, in the sense of being able to solve new problems, and exhibit backward transfer, meaning performance on previously encountered problems should increase after solving new ones. Ideally, the algorithm should be capable of transfer across similar input distributions (*perceptual transfer*), dissimilar input distributions and different input spaces (transfer of latent concepts, i.e., *latent transfer*), and to problems with few training examples (*few-shot transfer*).

Recent work (Valkov et al., 2018; Veniat et al., 2020; Ostapenko et al., 2021) has shown modular algorithms to be a promising approach to CL. These methods represent a neural network as a composition of reusable modules. During learning, the algorithms accumulate a library of diverse modules by solving the encountered problems in a sequence. Given a new problem, they seek to find the best composition of pre-trained and new modules as measured by the performance on a held-out dataset.

Modular methods can achieve plasticity using fresh modules, while using pre-trained modules to avoid catastrophic forgetting. However, *scalability* remains a key challenge in these methods, as the module "library" grows linearly in the number of problems, the set of all module compositions grows polynomially in the size of the library, and hence exponentially in the number of problems. Further, the evaluation of the quality of a composition requires the expensive training of its modules.

In this paper, we present a modular CL framework, called PICLE[1], that mitigates this challenge through a probabilistic search (Shahriari et al., 2015). The central insight is that searching over module compositions would be substantially cheaper if we could compute a composition's fitness, proportional to its final performance, without training the new modules in it. Accordingly, PICLE uses a probabilistic model to approximate the fitness of each composition.

The benefit of such a probabilistic model is that it combines prior knowledge about good module compositions with problem-specific information in a principled way. We introduce two different

---

[*]valkovl@mit.edu
[1]PICLE, pronounced "pickle", stands for **P**robabilistic, **LI**brary-based **C**ontinual **Le**arning.

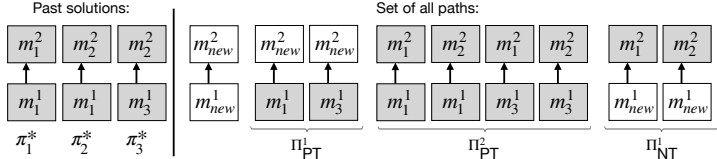

Figure 1: The set of all paths that a modular algorithm considers when solving the 4th problem in a sequence. The modular architecture has $L = 2$ layers. The shaded modules are re-used from previous problems. The library comprises all pre-trained modules: $\mathcal{L} = \{m_1^1, m_3^1, m_1^2, m_2^2\}$. Paths in $\Pi_{\text{PT}}^1$ (Section 4) select a pre-trained module for the first layer, enabling perceptual transfer. Paths in $\Pi_{\text{PT}}^2$ reuse modules in both layers. They can perform few-shot transfer since they only require a few examples (to select the correct path). Paths in $\Pi_{\text{NT}}^1$ (Section 5) achieve latent transfer by reusing a module in the second layer, allowing applications to new input domains.

probabilistic models, specialized for different types of transfer: one for perceptual and few-shot transfer that is focused on modelling the distribution over input activations to each module (Section 4); one for latent transfer which is based on a Gaussian process to capture the idea that similar compositions of modules should have similar performance (Section 5). We make use of both models to create a scalable modular CL algorithm with constant training requirements, i.e., it evaluates a constant number of compositions and trains networks with a fixed size for each problem.

We evaluate PICLE on the popular CTrL benchmark suite (Veniat et al., 2020), as well as a new extension of CTrL, which we call BELL, which introduces compositional tasks. Comparing to a wide range of approaches, our experiments demonstrate that PICLE is the first modular CL algorithm to achieve perceptual, few-shot and latent transfer while scaling well to large search spaces, outperforming previous state-of-the-art modular CL approaches on long problem sequences.

## 2 BACKGROUND

A CL algorithm aims to solve a sequence of problems, usually provided one at a time. We consider the supervised setting, in which each problem is characterized by a training set $(\mathbf{X}^{\text{tr}}, \mathbf{Y}^{\text{tr}})$ and a validation set $(\mathbf{X}^{\text{val}}, \mathbf{Y}^{\text{val}})$. The algorithm's goal is to transfer knowledge between the problems. It is considered *scalable* if both its memory and computational requirements scale sub-linearly with the number of solved problems (Veniat et al., 2020).

Modular approaches to CL (Valkov et al., 2018; Veniat et al., 2020; Ostapenko et al., 2021) represent a deep neural network as a composition of modules, each of which is a parameterized nonlinear transformation. Transfer learning is accomplished by reusing modules from previous learning problems. Each composition is represented by *a path*: a sequence of modules $\pi = (m^i)_{i=1}^L$ that defines a function composition $m^L \circ ... \circ m^2 \circ m^1$. We refer to each element of the path as a *layer*. After solving $t-1$ problems, modular algorithms accumulate a library of frozen previously trained modules. We denote the set of all pre-trained modules in the library for layer $i$ as $\mathcal{L}^i$.

Given a new learning problem, one can construct different modular neural networks by searching over the set of paths (Fig. 1). For each layer $i$, a path selects either a pre-trained module from the library $\mathcal{L}^i$ or introduces a new one $m_{\text{new}}^i$ that needs to be trained from scratch. In order to find the best path, a modular algorithm assesses different paths by training the resulting neural networks and evaluating their *final validation performance* on the validation set, denoted as $f$. Thus, evaluating each path is costly, and furthermore, the number of paths grows rapidly with the size of the library.

## 3 A PROBABILISTIC FRAMEWORK FOR MODULAR CL

PICLE is based on a probabilistic search over paths. For each path, we seek to compute, without training the path, a proxy for its validation accuracy. We compute this proxy, called *fitness*, as a posterior distribution over the choice of pre-trained modules specified by a path, given the problem's data. In doing so, we leverage both prior and problem-specific knowledge.

We distinguish between two types of paths that lead to different types of transfer and require different probabilistic models: (i) Paths that enable *perceptual transfer* (PT) between similar input

distributions, by using pre-existing modules in the first $l$ layers and new modules elsewhere (Section 4); and (ii) paths that allow *latent transfer* or non-perceptual transfer (NT) between problems with different input distributions or different input spaces, by using pre-existing modules in the last $l$ layers and new modules before them. The two kinds of paths are illustrated in Fig. 1.

For PT paths, designing a probabilistic model is easier, as we can evaluate the pre-trained modules on data from the new problem. For NT paths, this is impossible because the initial modules in the network are untrained. Hence, we define a different probabilistic model, which employs a distance in function space across compositions of pre-trained modules.

---

**Algorithm 1:** PICLE

1   $\mathcal{L} \leftarrow (), \boldsymbol{\pi}^* \leftarrow ()$ // Empty library, no solutions.
2   **for** *each problem $t$ with data* $\mathbf{X}^{tr}, \mathbf{Y}^{tr}, \mathbf{X}^{val}, \mathbf{Y}^{val}$ **do**
3     $\pi_{\text{SA}}^* \leftarrow (m_{\text{new}}^1, ..., m_{\text{new}}^L)$ // Evaluate a fully randomly initialized network.
4     $f_{\text{SA}}^* \leftarrow \text{TRAINANDEVALUATE}(\pi_{\text{SA}}^*, \mathbf{X}^{tr}, \mathbf{Y}^{tr}, \mathbf{X}^{val}, \mathbf{Y}^{val})$
5     $\pi_{\text{PT}}^*, f_{\text{PT}}^* \leftarrow \text{FINDBESTPTPATH}(\mathcal{L}, \mathbf{X}^{tr}, \mathbf{Y}^{tr}, \mathbf{X}^{val}, \mathbf{Y}^{val})$
6     $\pi_{\text{NT}}^*, f_{\text{NT}}^* \leftarrow \text{FINDBESTNTPATH}(\mathcal{L}, \boldsymbol{\pi}^*, \mathbf{X}^{tr}, \mathbf{Y}^{tr}, \mathbf{X}^{val}, \mathbf{Y}^{val})$
7     $\pi^* \leftarrow \pi_j^*$ where $j = \arg\max_{j' \in \{\text{SA}, \text{PT}, \text{NT}\}} f_{j'}^*$ // select the best performing path
8     $\text{UPDATELIBRARY}(\mathcal{L}, \pi^*)$ ; $\text{APPEND}(\boldsymbol{\pi}^*, \pi^*)$

---

Our overall framework is shown in Algorithm 1. For each new learning problem, PICLE first evaluates a fully randomly initialized, i.e. a *standalone* (SA), model in case no pre-trained module is useful. Afterwards, it searches for the best path, combining the PT and NT search strategies. Finally, the UPDATELIBRARY function adds modules from the best path to the library for future problems.

## 4   SCALABLE PERCEPTUAL TRANSFER AND FEW-SHOT TRANSFER

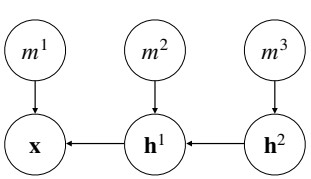

Figure 2: Our probabilistic model for a PT path with three pre-trained modules, $m^1, m^2, m^3$ and their respective inputs $\mathbf{x}, \mathbf{h}^1$ and $\mathbf{h}^2$.

Our method searches through paths, which achieve perceptual transfer (PT paths) by reusing modules in the first $l$ layers (Fig. 1). This includes cases where *all* layers are reused, potentially combining modules from different problems. Such a path can perform *few-shot transfer*.

The insight here is to select each pre-trained module so that its distribution over inputs on the new problem is similar to the input distribution it was trained on, minimizing distribution shift. Accordingly, we define a probabilistic model over the choice of pre-trained modules and their inputs, and introduce a search strategy that searches for the PT path with the highest probability.

The set of PT paths with prefix-length $\ell$ is the Cartesian product of library items $\mathcal{L}^i$ for the first $i \in \{1, ..., \ell\}$ layers and new modules in the other layers: $\Pi_{\text{PT}}^\ell = \mathcal{L}^1 \times ... \times \mathcal{L}^\ell \times \{m_{\text{new}}^{\ell+1}\} \times ... \times \{m_{\text{new}}^L\}$. This set grows polynomially with the size of the library, making a naive search strategy inapplicable.

Instead, we introduce a strategy based on a generative model of the input $\mathbf{x}$, the choice of pre-trained modules $m^1 \ldots m^\ell$, and the latent activations $\mathbf{h}^1 \ldots \mathbf{h}^{\ell-1}$ after each pretrained module, i.e., $\mathbf{h}^{j+1} = m^{j+1}(\mathbf{h}_j)$. We imagine sampling the *highest-level* activation $\mathbf{h}^{\ell-1}$ first, and then sequentially sampling each lower-level activation $\mathbf{h}^j$ to satisfy the constraint $\mathbf{h}^{j+1} = m^{j+1}(\mathbf{h}^j)$. This process is illustrated by the graphical model in Figure 2. That is, we model the joint distribution as

$$p(m^1, ..., m^\ell, \mathbf{x}, \mathbf{h}^1..., \mathbf{h}^{\ell-1}) = p(\mathbf{x}|\mathbf{h}^1, m^1) \left( \prod_{i=1}^{\ell-2} p(\mathbf{h}^i | \mathbf{h}^{i+1}, m^{i+1}) \right) p(\mathbf{h}^{\ell-1}|m^\ell) \prod_{i=1}^{\ell} p(m^i). \tag{1}$$

This allows us to infer the distribution over pre-trained modules, given the observed inputs. Because $\mathbf{h}^{j+1} = m^{j+1}(\mathbf{h}^j)$, we can marginalize out the activations and express the posterior distribution over pre-trained modules using terms we can compute, namely, a prior $p(m^i)$ over pre-trained modules

---

**Algorithm 2:** FINDBESTPTPATH: Searching through perceptual-transfer paths

---

**Input:** A library $\mathcal{L}$ of pre-trained modules.
**Input:** The training data $(\mathbf{X}^{\text{tr}}, \mathbf{Y}^{\text{tr}})$ and validation data $(\mathbf{X}^{\text{val}}, \mathbf{Y}^{\text{val}})$ for the new problem.

1  $\boldsymbol{\pi} \leftarrow (\,), \mathbf{f} \leftarrow (\,);$ `// evaluated paths and their final validation performances`
2  $\mathbf{m} \leftarrow (\,)$ `// selected pre-trained modules`
3  **for** $\ell \leftarrow 1$ **to** $L$ **do**
4  $\quad m^\ell \leftarrow \arg\max_{m \in \mathcal{L}^\ell} p\left(\mathbf{m} \oplus (m) \,|\, \mathbf{X}^{\text{tr}}\right)$
5  $\quad \pi \leftarrow \mathbf{m} \oplus (m^\ell) \oplus (m^{\ell+1}_{\text{new}}, ..., m^L_{\text{new}})$ `// ⊕ denotes sequence concatenation`
6  $\quad f \leftarrow$ TRAINANDEVALUATE$(\pi, \mathbf{X}^{\text{tr}}, \mathbf{Y}^{\text{tr}}, \mathbf{X}^{\text{val}}, \mathbf{Y}^{\text{val}})$
7  $\quad$ APPEND$(\boldsymbol{\pi}, \pi);$ APPEND$(\mathbf{f}, f);$ APPEND$(\mathbf{m}, m^\ell)$
8  $i \leftarrow \arg\max_{i'} \mathbf{f}[i']$
9  **return** $\boldsymbol{\pi}[i], \mathbf{f}[i]$ `// returning best path and its final validation performance`

---

and an approximation of the input distribution $p(\mathbf{h}^{i-1}|m^i)$ for each pre-trained module:

$$p(m^1, ..., m^\ell \,|\, \mathbf{x}) \propto p(m^1, ..., m^\ell, \mathbf{x}) = p(\mathbf{x}|m^1) \left(\prod_{i=2}^{\ell-1} \frac{p(\mathbf{h}^{i-1}|m^i)}{\sum_{m \in \mathcal{L}_i} p(\mathbf{h}^{i-1}|m)p(m)}\right) \prod_{i=1}^{\ell} p(m^i). \quad (2)$$

We consider that a pre-trained module's competence can be measured by the accuracy it achieved on the problem it was originally trained to solve. Accordingly, our prior $p(m^i)$ uses softmax to order modules by their original accuracy (see Appendix E). To approximate the training input distribution $p(\mathbf{h}^{i-1}|m^i)$ of $m^i$, we first choose a low-rank Gaussian distribution. We estimate its parameters using the set of hidden activations $\mathbf{h}^{i-1}_1 \ldots \mathbf{h}^{i-1}_N \in \mathbb{R}^v$ that were provided as inputs to $m^i$ during its training. We use a random projection (Johnson, 1984) to reduce each hidden activation's dimensionality to $k \ll v$. Next, we approximate the distribution in the low-dimensional space by a multivariate Gaussian, by computing the sample mean and covariance of the projected data samples. This greatly reduces the number of parameters required to approximate the resulting distribution from $v + v^2$ to $k + k^2$. Surprisingly, our ablation experiments (Appendix I) suggest that this approximation also leads to a more reliable search algorithm.

**Search Strategy and Scalablity.** Our overall algorithm (Algorithm 2) uses a greedy search to find a PT path of prefix-length $\ell \in \{1 \ldots L\}$ with the highest posterior probability $p(m_1 \ldots m_\ell \,|\, \mathbf{X}^{\text{tr}})$. At each iteration $\ell$, the algorithm extends the current prefix with a library module $m_\ell \in \mathcal{L}^\ell$ which leads to the highest posterior probability. This leads to evaluating one path for each prefix-length $\ell \in \{1 \ldots L\}$, and we return the single path with best validation performance. Importantly, our greedy search strategy scales well in the number of problems: it trains $L$ networks each of which contains $L$ modules, so the training requirements are constant in the size of the library.

## 5 SCALABLE LATENT TRANSFER

In the latent transfer setting, the search space is the set of *latent transfer (NT) paths* (Fig. 1). These are paths where the first $L - \ell$ modules are randomly initialised and the final $\ell$ modules, referred to as the *pre-trained suffix*, are selected from the library. The set of all NT paths of *suffix-length* $\ell$ is: $\Pi^\ell_{\text{NT}} = \{m^1_{\text{new}}\} \times ... \times \{m^{L-\ell}_{\text{new}}\} \times \mathcal{L}^{L-\ell+1} \times ... \times \mathcal{L}^L$, which grows exponentially in $\ell$.

For efficient search, we need a model that captures how good an NT path is at predicting the correct outputs after training. This is difficult as the distribution over inputs for the pre-trained modules depends on the earlier modules that we wish to avoid training. We overcome this challenge by defining a model that directly approximates an NT path's generalization performance after training. We note that the final $\ell$ layers of an NT path define a function from input activations to output activations. This allows us to employ a similarity metric between NT paths in function space. Specifically, we use a Gaussian process based on the intuition that two paths that define similar functions are likely to perform similarly when stacked with new random modules and fine-tuned on a new dataset.

Our model defines a distribution over the pre-trained modules of an NT path. For an NT path $\pi \in \Pi^\ell_{\text{NT}}$, we specify a prior over its pre-trained modules $p(m^{L-\ell+1}, ..., m^L)$ and approximate the final validation performance that using these modules would achieve after training:

$p(\mathbf{Y}^{\text{val}}|\mathbf{X}^{\text{val}}, m^{L-\ell+1}, ..., m^L)$. Consequently, we express the posterior over an NT path's pre-trained modules given the validation dataset as:

$$p(m^{L-\ell+1}, ..., m^L|\mathbf{X}^{\text{val}}, \mathbf{Y}^{\text{val}}) \propto p(\mathbf{Y}^{\text{val}}|\mathbf{X}^{\text{val}}, m^{L-\ell+1}, ..., m^L)p(m^{L-\ell+1}, ..., m^L). \quad (3)$$

Our prior states that pre-trained modules in the path must have all been used together to solve a previous problem. Therefore, $p(m^{L-\ell+1}, ..., m^L) \propto 1$ if and only if the modules $m^{L-\ell+1}, ..., m^L$ are a suffix of a previous solution characterised by some path $\pi^*$. This prior, used by Veniat et al. (2020) for perceptual transfer, reflects our assumption that using novel combination of modules for latent transfer is unnecessary for our sequences.

To approximate the final validation performance of an NT path $p(\mathbf{Y}^{\text{val}}|\mathbf{X}^{\text{val}}, m^{L-\ell+1}, ..., m^L)$, we first note that it is a function of the pre-trained suffix $\lambda = (m^{L-\ell+1}, ..., m^L)$, since the validation data is fixed for a given problem, and denote it as $f(\lambda) = p(\mathbf{Y}^{\text{val}}|\mathbf{X}^{\text{val}}, \lambda)$. Next, we approximate it by putting a Gaussian Process (GP) prior with kernel $\kappa$ on it: $f(\lambda) \sim GP(\mathbf{0}, \kappa(\lambda, \lambda'))$. Here, $\lambda'$ denotes another pre-trained suffix of the same length. To define a kernel function $\kappa$, we note that the pre-trained modules for an NT path, $m^{L-\ell+1}, ..., m^L$ compute a function $m^{L-\ell+1} \circ ... \circ m^L$. We hypothesise that if two NT paths' pre-trained suffixes compute similar functions, then their final validation performances will be also be similar. Accordingly, we capture this by using the squared exponential kernel function $\kappa(\lambda, \lambda') = \sigma^2 \exp\left\{-d(\lambda, \lambda')^2/(2\gamma^2)\right\}$, where $d$ is the distance between two functions and $\sigma$ and $\gamma$ are the kernel hyperparameters which are fit to maximize the marginal likelihood of a GP's training data (Rasmussen & Williams, 2006).

We compute the Euclidean distance $d$ between two functions (Appendix F) which we approximate using Monte Carlo integration with a set of inputs from the functions' common input space. For pre-trained suffixes of length $\ell$, we store a few hidden activations from the input distribution of each pre-trained module at the first pre-trained layer $L - \ell + 1$. Given a new problem, we create a set of function inputs by combining all the stored hidden activations. For other kernels, see Appendix J.

---

**Algorithm 3:** FINDBESTNTPATH: Searching through latent-transfer paths

**Input:** A library $\mathcal{L}$ of pre-trained modules.
**Input:** A list of paths $\boldsymbol{\pi}^*$ of solutions to previous problems.
**Input:** The training data $(\mathbf{X}^{\text{tr}}, \mathbf{Y}^{\text{tr}})$ and validation data $(\mathbf{X}^{\text{val}}, \mathbf{Y}^{\text{val}})$ for the new task.

1   $\boldsymbol{\pi} \leftarrow (), \mathbf{f} \leftarrow ();$ `// evaluated paths and their validation performance`
    `// Perform Bayesian Opt. for NT paths with` $\ell_{\min}$ `pre-trained modules:`
    `// Compute all suffixes of length` $\ell_{\min}$ `from previous solutions,` $\boldsymbol{\pi}^*$`.`
2   $\boldsymbol{\lambda} \leftarrow (\pi'[L - \ell_{\min} + 1 : L] \text{ for } \pi' \in \boldsymbol{\pi}^*)$
3   $\boldsymbol{\lambda}' \leftarrow ()$ `// evaluated suffixes`
    `// Find the most relevant previous solution` $\pi'$`.`
4   **for** $iter \leftarrow 1$ **to** $min(c, len(\boldsymbol{\pi}^*))$ **do**
5      $\lambda \leftarrow \arg\max_{\lambda \in \boldsymbol{\lambda}} \text{UCB}(p_{\text{GP}}(f|\lambda, \boldsymbol{\lambda}', \mathbf{f}))$
6      $\pi \leftarrow (m^1_{\text{new}}, ..., m^{L-\ell_{\min}}_{\text{new}}) \oplus \lambda$ `//` $\oplus$ `denotes sequence concatenation`
7      $f \leftarrow \text{TRAINANDEVALUATE}(\pi, \mathbf{X}^{\text{tr}}, \mathbf{Y}^{\text{tr}}, \mathbf{X}^{\text{val}}, \mathbf{Y}^{\text{val}})$
8      APPEND($\boldsymbol{\pi}, \pi$); APPEND($\boldsymbol{\lambda}', \lambda$); APPEND($\mathbf{f}, f$)
9   $i \leftarrow \arg\max_i \mathbf{f}[i]$
10   $\pi' \leftarrow$ the path in $\boldsymbol{\pi}^*$ that has $\boldsymbol{\lambda}'[i] = \pi'[L - \ell_{\min} + 1 : L]$
     `// Find the suffix of` $\pi'$ `that has the best transfer:`
11   **for** $\ell \leftarrow L - \ell_{\min}$ **to** $2$ **do**
12      $\pi \leftarrow (m^1_{\text{new}}, ..., m^{\ell-1}_{\text{new}}) \oplus \pi'[\ell : L]$
13      $f \leftarrow \text{TRAINANDEVALUATE}(\pi, \mathbf{X}^{\text{tr}}, \mathbf{Y}^{\text{tr}}, \mathbf{X}^{\text{val}}, \mathbf{Y}^{\text{val}})$
14      APPEND($\boldsymbol{\pi}, \pi$); APPEND($\mathbf{f}, f$)
15   $i \leftarrow \arg\max_{i'} \mathbf{f}[i']$
16   **return** $\boldsymbol{\pi}[i], \mathbf{f}[i]$ `// return best path and its final validation performance`

---

**Search Strategy and Scalability.** We can use our probabilistic model to define a scalable search strategy (Algorithm 3) over a set of NT paths. Due to our prior, we only consider NT paths that have the same suffix as a previous solution. Denote by $\boldsymbol{\pi}^*$ the set of previous solutions, i.e., the paths that were used to solve previous learning problems. Our method first searches for the previous solution $\pi' \in \boldsymbol{\pi}^*$ that is most relevant to the current learning task, using Bayesian optimization. Then we search over the suffixes of $\pi'$ in particular, and choose exactly how many of its modules to reuse.

Lines 4-8 of our algorithm use our probabilistic model to search for the most relevant previous solution $\pi'$. We assess the relevance of a solution by trying to transfer its last $\ell_{\min}$ modules, and choosing the solution which leads to the best performance. The value $\ell_{\min}$ is a hyperparameter, which we choose to be the minimum number of pre-trained modules needed to obtain increased generalisation performance. We evaluate up to $c$ paths, where $c$ is a hyperparameter. At each step, we have a sequence of already evaluated NT paths $\boldsymbol{\pi}$, their pre-trained suffixes $\boldsymbol{\lambda'}$, and their final validation performance $\mathbf{f}$. We use a GP and combine its predictive distribution $p_{\mathrm{GP}}$ with the Upper Confidence Bound (UCB) Srinivas et al. (2009) acquisition function, to predict each unevaluated path's final validation performance. This search results in the most relevant previous solution $\pi'$. Finally, in lines 11-14, we evaluate NT paths created by transferring a different number of the last layers of $\pi'$, to see if re-using more layers leads to further improvement.

Overall, this search through NT paths requires training and evaluating up to a constant number of paths, concretely up to $c + L - \ell_{\min} - 1$, in the size of the library, with each path specifying a with a constant number of $L$ modules, allowing it to scale well with the number of problems.

# 6    EXPERIMENTS

We compare our algorithm against several competitive modular CL baselines: RS, which randomly selects from the set of all paths (shown to be a competitive baseline in high-dimensional search spaces and in neural architecture search (Li & Talwalkar, 2020)); HOUDINI (Valkov et al., 2018), with a fixed neural architecture to keep the results comparable; MNTDP-D (Veniat et al., 2020) which is a scalable modular CL algorithm for perceptual transfer; LMC (Ostapenko et al., 2021) which can achieve different transfer properties but has limited scalability. For completeness, we also compare to a standalone (SA) baseline which always trains a new network for every problem; a CL algorithm based on parameter regularisation, online EWC (O-EWC) (Schwarz et al., 2018; Chaudhry et al., 2018); and one based on experience replay, ER (Chaudhry et al., 2019).

Our hyperparameters are listed in Appendix G. For each baseline, we assess the performance on a held-out test dataset. To measure transfer learning performance, we report amount of forward transfer on the last problem, $Tr^{-1}$, computed as the difference in accuracy, compared to the standalone baseline (see Eq. 5). We also report the average forgetting $\mathcal{F}$ across all sequences, measured by the difference between the accuracy achieved on each problem at the end of the sequence and initially (see Eq. 6). Additionally, for completeness, we report the average accuracy of the final model across all problems, $\mathcal{A}$ (see Eq. 4); this averages over problems earlier in the sequence, for which no transfer is possible by design, so it is less helpful for comparing transfer learning approaches.

**Compositional Benchmarks.** The CTrL benchmark (Veniat et al., 2020) has been the most popular for evaluating recent modular CL approaches (Veniat et al., 2020; Ostapenko et al., 2021). It defines sequences of learning problems, each sequence being designed to evaluate a particular CL desideratum. We introduce a new, compositional extension of CTrL, which we call BELL. Each problem in BELL is a binary classification task on a pair of images, which composes an image classification task with a pattern recognition task; for example, "do both images depict a number less than 5". This allows us to sample a larger variety of random problems by choosing different classification and pattern recognition tasks. Most sequences contain 6 problems, except for a sequence of length 60 that tests scalability. Our neural architecture has 8 modules, which leads to a large search space (e.g., for the 6th problem, there are $\mathcal{O}(6^8)$ possible paths). Details on the problems, sequences, neural architecture and training procedure can be found in Appendix H. For each sequence, we create 3 versions by sampling different classification tasks, and averaging the metrics over these 3 versions.

*Results.* Overall, PICLE outperforms the other methods, achieving, on average across the short sequences, $+2.7$ higher final accuracy and $+9.02$ higher transfer on the last problem, compared to the second-best algorithm MNTDP-D (Tables 1 and 2). PICLE and MNTDP-D achieve similar performances on sequences evaluating their shared CL properties (perceptual transfer, plasticity, stability), but PICLE also demonstrates few-shot transfer and latent transfer. LMC performed worse than standalone on these sequences despite our efforts to adjust it to this setting, so we do not report it. Forgetting impacts O-EWC and ER's final accuracy but is avoided by modular algorithms.

First, we evaluate the methods' *perceptual transfer capabilities* via sequences $S^-$, $S^{\mathrm{out}}$, $S^{\mathrm{out}*}$, $S^{\mathrm{out}**}$. These are listed in increasing order of difficulty; the last problem in the sequence has an input

Table 1: Results on the compositional benchmarks from BELL which assess perceptual transfer ($S^-$, $S^{\text{out}}$, $S^{\text{out*}}$, $S^{\text{out**}}$), few-shot transfer ($S^{\text{few}}$), plasticity ($S^{\text{pl}}$) and backward transfer ($S^+$).

| | | SA | O-EWC | ER | RS | HOUDINI | MNTDP-D | PICLE |
|---|---|---|---|---|---|---|---|---|
| | $S^{\text{few}}$ | 75.47 | 50.92 | 63.31 | 78.14 | 80.82 | 82.18 | **88.12** |
| | $S^{\text{out}}$ | 74.25 | 56.98 | 60.58 | 76.16 | 74.40 | 77.95 | **78.15** |
| | $S^{\text{out*}}$ | 72.27 | 56.72 | 59.5 | 73.39 | 72.27 | 75.48 | **75.72** |
| | $S^{\text{out**}}$ | 71.51 | 55.74 | 59.27 | 73.85 | 71.75 | 73.71 | **75.73** |
| $\mathcal{A}$ | $S^{\text{pl}}$ | 93.61 | 58.87 | 64.27 | 93.63 | 93.61 | 93.72 | **93.79** |
| | $S^-$ | 73.88 | 58.07 | 65.75 | 76.67 | 79.59 | 81.67 | **81.92** |
| | $S^+$ | 73.61 | 59.96 | 67.34 | **75.08** | 73.61 | 74.54 | 74.49 |
| | **Avg.** | 76.37 | 56.75 | 62.86 | 78.13 | 78.01 | 79.89 | **81.13** |
| $\mathcal{F}$ | **Avg.** | 0. | -17.99 | -11.11 | 0. | 0. | 0. | **0.** |
| | $S^{\text{few}}$ | 0. | 4.33 | 1.5 | 5.87 | 4.54 | 11.42 | **46.07** |
| | $S^{\text{out}}$ | 0. | 1.57 | -13.27 | 5.64 | 0. | **15.41** | **15.41** |
| | $S^{\text{out*}}$ | 0. | 0.29 | -7.91 | 0.43 | 0. | **12.53** | **12.53** |
| $Tr^{-1}$ | $S^{\text{out**}}$ | 0. | 4.21 | -8.1 | 4.61 | 1.46 | 1.74 | **12.04** |
| | $S^{\text{pl}}$ | 0. | -22.34 | -1.09 | 0. | 0. | **00.20** | **00.20** |
| | $S^-$ | 0. | 4.8 | 8.92 | 17.22 | 34.27 | **34.29** | **34.29** |
| | $S^+$ | 0. | -2.63 | -0.57 | 0. | 0. | 0. | **0.** |
| | **Avg.** | 0. | -1.4 | -2.93 | 4.82 | 5.75 | 10.8 | **17.22** |

Table 2: Results on compositional benchmarks from BELL which assess latent transfer.

| | | SA | O-EWC | ER | RS | HOUDINI | MNTDP-D | PT-only | NT-only | PICLE |
|---|---|---|---|---|---|---|---|---|---|---|
| | $S^{\text{in}}$ | 89.01 | 57.78 | 62.68 | 90.85 | 89.32 | 90.62 | 90.26 | 92.20 | **92.82** |
| $\mathcal{A}$ | $S^{\text{sp}}$ | 87.94 | 60.04 | 64.97 | 92.22 | **92.99** | 87.94 | 87.92 | 91.92 | 91.93 |
| | **Avg.** | 88.48 | 58.91 | 63.83 | 91.54 | 91.16 | 89.28 | 89.09 | 92.06 | **92.38** |
| $\mathcal{F}$ | **Avg.** | 0. | -24.55 | -25.73 | 0. | 0. | 0. | 0. | 0. | **0.** |
| | $S^{\text{in}}$ | 0. | -0.77 | 6.17 | 1.81 | 11.04 | 9.70 | 7.61 | 18.89 | **22.28** |
| $Tr^{-1}$ | $S^{\text{sp}}$ | 0. | -2.59 | -3.56 | 25.68 | **30.27** | 0. | 0. | 23.65 | 23.65 |
| | **Avg.** | 0. | -1.68 | 1.31 | 13.75 | 20.66 | 4.85 | 3.805 | 21.27 | **22.97** |

domain similar to one of the previous problems but is represented by a low number of data points. We consider the amount of transfer achieved in the last problem, $Tr^{-1}$ of Table 1. HOUDINI fails to scale to the large search space, only achieving transfer on the easiest sequence $S^-$. As expected, we find that MNTDP-D and PICLE are equally capable of perceptual transfer and both perform significantly better than other methods. However, on $S^{\text{out**}}$, we find that PICLE achieves $+10.33$ higher transfer, made possible by the performance-based prior we defined in Section 4 for PT paths.

Second, *few-shot transfer* is evaluated by the $S^{\text{few}}$ sequence (Table 1) in which the last problem is represented by a few examples and shares an input domain of one of the past problems, and the two-dimensional pattern of another. This requires an algorithm to recompose previously acquired knowledge. HOUDINI demonstrates low amount of transfer due to the large search space. MNTDP-D selects the correct modules to process the input domain but is unable to compose them with modules from another solution, leading to sub-optimal performance. Finally, PICLE successfully achieves few-shot transfer, attaining $+34.65$ higher transfer ($Tr^{-1}(S^{\text{few}})$) than MNTDP-D.

Third, *latent transfer* is evaluated by two sequences (Table 2): $S^{\text{in}}$ and $S^{\text{sp}}$. In both, the last problem shares the two-dimensional pattern of the first problem, but has a new image input domain ($S^{\text{in}}$) or a new input space ($S^{\text{sp}}$). MNTDP-D does not achieve latent transfer: it transfers some perceptual knowledge on $S^{\text{in}}$ and fails on $S^{\text{sp}}$. HOUDINI performs similarly to MNTDP-D on $S^{\text{in}}$ but achieves the largest amount of latent transfer on $S^{\text{sp}}$. In $S^{sp}$, the different input space necessitates a different modular architecture for the first 5 modules, resulting in a much smaller search space, $\mathcal{O}(6^3 = 216)$. This allows non-scalable approaches, namely RS and HOUDINI, to also be effective on this sequence. Finally, PICLE demonstrates latent transfer on both sequences, outperforming MNTDP-D and transferring $+14.67$ ($Tr^{-1}(S^{\text{in}})$) and $+23.65$ ($Tr^{-1}(S^{\text{sp}})$) compared to the PT-only ablation.

An algorithm's *plasticity*, i.e. ability to acquire new knowledge, is evaluated by $S^{\text{pl}}$ in which all problems are different. All modular algorithms demonstrate plasticity, as they can always introduce a new set of modules, achieving similar final accuracies $\mathcal{A}(S^{\text{pl}})$ (Table 1). The last short sequence $S^+$ evaluates the backward transfer by having the last problem be the same as the first one but with

Table 3: Results on CTrL sequences which assess perceptual transfer ($S^-$, $S^{\text{out}}$), plasticity ($S^{\text{pl}}$), backward transfer ($S^+$) and latent transfer $S^{\text{in}}$. Standard deviations appear in Appendix K.

| | | SA | O-EWC | ER | RS | HOUDINI | LMC | MNTDP-D | PT-only | NT-only | PICLE |
|---|---|---|---|---|---|---|---|---|---|---|---|
| | $S^{\text{in}}$ | 65.43 | 24.49 | 46.4 | 65.41 | 64.59 | **69.45** | 66.23 | 65.86 | 68.18 | 68.34 |
| | $S^{\text{out}}$ | 63.11 | 46.93 | 56.56 | 63.77 | 66.54 | 65.72 | **66.78** | **66.78** | - | **66.78** |
| | $S^{\text{pl}}$ | 63.97 | 43.64 | 58.82 | 63.72 | 63.89 | 62.31 | 64.56 | **64.67** | - | **64.67** |
| $\mathcal{A}$ | $S^-$ | 62.96 | 48.3 | 54.87 | 64.07 | 67.16 | 65.77 | **67.24** | 67.24 | - | 67.24 |
| | $S^+$ | 63.04 | 45.93 | 56.8 | 63.15 | 62.83 | 59.68 | **63.11** | 62.99 | - | 62.99 |
| | **Avg.** | 63.70 | 41.86 | 54.69 | 64.02 | 65. | 64.59 | 65.58 | 65.51 | - | **66.00** |
| $\mathcal{F}$ | **Avg.** | 0. | -20.65 | -4.78 | 0. | 0. | -0.66 | 0. | 0. | 0. | **0.** |
| | $S^{\text{in}}$ | 0. | 3.79 | -41.13 | -0.53 | -5.03 | **17.41** | -0.63 | 1.36 | 16.26 | 16.26 |
| | $S^{\text{out}}$ | 0. | 6.75 | 2.88 | 2.64 | **17.46** | 16.79 | 17.06 | 17.06 | - | 17.06 |
| $Tr^{-1}$ | $S^{\text{pl}}$ | 0. | -13.89 | -6.5 | **0.** | **0.** | -10.79 | **0.** | **0.** | - | **0.** |
| | $S^-$ | 0. | 6.7 | 2.42 | 4.09 | **20.74** | 18.38 | **20.74** | 20.74 | - | 20.74 |
| | $S^+$ | 0. | -14.57 | -7.52 | 0. | 0. | -3.0 | 0. | 0. | - | **0.** |
| | **Avg.** | 0. | -2.24 | -9.97 | 1.24 | 6.63 | 7.76 | 7.43 | 7.83 | - | **10.81** |

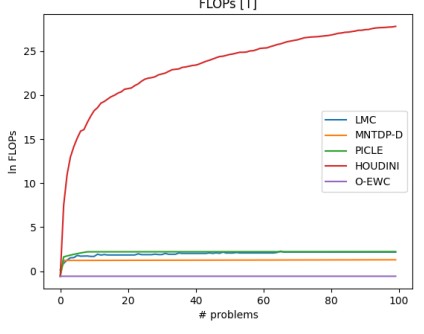

(a) A logarithmic scale of the number of FLOPs required for a fixed number of updates, per problem (in trillions)

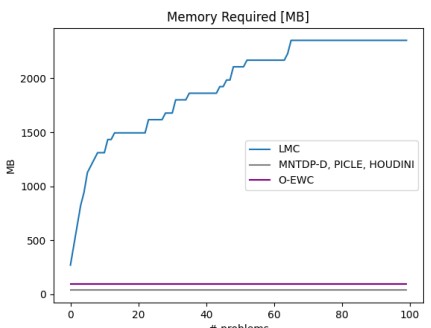

(b) The maximum memory required by each algorithm for each problem.

Figure 3: Resource requirements for CTrL's $S^{\text{long}}$.

a larger dataset. All modular algorithms showed inability to attain backward transfer, due to their pre-trained modules being frozen, thus, achieving similar final accuracies $\mathcal{A}(S^{\text{few}})$.

Finally, to measure scalability, we define $S^{\text{long}}$ which contains 60 randomly-selected compositional problems. We compare to MNTDP-D as it was the only other method able to do well in the large search space of the short sequences. PT-only achieved $+7.37$ higher average accuracy than the standalone baseline demonstrating its ability to achieve perceptual transfer on a long sequence of problems. MNTDP-D achieved $+8.83$ higher average accuracy than SA which confirmed its scalability. PICLE performed the best, attaining $+12.25$ higher average accuracy than SA. This shows that PICLE can successfully attain perceptual and latent transfer across a long problem sequence.

**CTrL benchmarks.** The CTrL benchmark suite defines fewer sequences to evaluate different CL properties. Namely, they specify $S^{\text{pl}}$, $S^+$, $S^-$, $S^{\text{out}}$, $S^{\text{sp}}$ which are defined similarly to ours. In contrast, the sequences are over multi-class classification tasks of coloured images from different domains. They also use a different modular architecture based on ResNet18 (He et al., 2016), which is more complex than the architecture used in the previous benchmarks. Our experimental setup, detailed in Appendix K, mirrors the one used in Ostapenko et al. (2021), except that we are averaging over three random seeds instead of a single one, all using the same data. Each network composes 5 modules, leading to a smaller search space on the short sequences – $\mathcal{O}(6^5)$.

Our results (Table 3) show similar performance of the different modular CL algorithms on the short sequences, with PICLE achieving the highest average performance. The smaller search space alleviates HOUDINI's and LMC's scalability issues. As a result, PICLE, HOUDINI, MNTDP-D and LMC demonstrate perceptual transfer ($S^-$, $S^{\text{out}}$), plasticity ($S^{\text{pl}}$) and stability ($S^+$). However, latent transfer ($S^{in}$) is exhibited by PICLE, HOUDINI and LMC, but not by MNTDP-D. Overall, the demonstrated properties are consistent with the ones observed on the compositional benchmarks. We also find PICLE to be effective on more complex input domains and neural architectures.

CTrL also specifies $S^{\text{long}}$, which has 100 problems. Each problem is constructed by first sampling one of 6 possible datasets (including CIFAR100 (Krizhevsky et al., 2009)), then 5 classes from that dataset at random. The 100 tasks represent a varying degree of task-relatedness and distribution shifts, making it representative of the typical CL benchmarks where one introduces new tasks (task-incremental), new classes (class-incremental) or new domains (domain-incremental). This more challenging sequence is designed to evaluate an algorithm's scalability. On it, PICLE achieved the highest average accuracy (**69.65**), compared to SA (55.04), MNTDP-D (65.64) and LMC (64.49). This demonstrates PICLE's effectiveness on large search spaces. Moreover, Figure 3 plots the computational and memory demands of modular CL algorithms on this sequence. It can be seen that LMC's memory and HOUDINI's computational requirements scale poorly with the number of problems, while MNTDP-D and PICLE's requirements grow slowly with the number of solved problems.

These results are summarised in Table 4. MNTDP-D and LMC are closest to PICLE in performance, but compared to MNTDP-D, PICLE is capable of latent and few-shot transfer and compared to LMC, PICLE has superior scalability. Our results demonstrate that PICLE is the first modular CL algorithm to achieve perceptual, few-shot and latent transfer while scaling to long problem sequences.

# 7 RELATED WORK

Table 4: Observed CL properties of modular CL approaches.

|  | RS | HOUDINI | LMC | MNTDP-D | PICLE |
|---|---|---|---|---|---|
| Plasticity | ✓ | ✓ | ✓ | ✓ | ✓ |
| Stability | ✓ | ✓ | ✓ | ✓ | ✓ |
| Perceptual Tr | ✓ | ✓ | ✓ | ✓ | ✓ |
| Few-shot Tr | ✓ | ✓ | ✓ | X | ✓ |
| Latent Tr | ✓ | ✓ | ✓ | X | ✓ |
| Scalability | X | X | X | ✓ | ✓ |
| Backward Tr | X | X | X | X | X |

This work considers the task-aware, data-incremental (De Lange et al., 2021) supervised setting of continual learning. Other settings can involve overlapping problems (Farquhar & Gal, 2018) or reinforcement learning (Khetarpal et al., 2020). Our CL desiderata is derived from Valkov et al. (2018) and Veniat et al. (2020). Other lists (Schwarz et al., 2018; Hadsell et al., 2020; Delange et al., 2021) do not distinguish between different types of forward transfer.

Continual learning methods can be categorized into ones based on regularisation, replay, or a dynamic architecture (Parisi et al., 2019). The first two share the same parameters across all problems, which limits their capacity and, in turn, their plasticity (Kirkpatrick et al., 2017). Dynamic architecture methods can share different parameters by learning problem-specific parameter masks (Mallya & Lazebnik, 2018) or adding more parameters (Rusu et al., 2016). This category includes modular approaches that share and introduce new modules, allowing groups of parameters to be trained and always reused together. Modular approaches mainly differ in their search space and their search strategy. PathNet (Fernando et al., 2017) uses evolutionary search to search through paths that combine up to 4 modules per layer. Rajasegaran et al. (2019) use random search on the set of all paths. HOUDINI (Valkov et al., 2018) uses type-guided exhaustive search on the set of all possible modular architectures and all paths. This method can attain the three types of forward transfer, but does not scale to large search spaces.

MNTDP-D (Veniat et al., 2020) is a scalable approach which uses a k-NN model to find the closest previous solution, and then restricts its search space to perceptual transfer paths derived from that solution. Similarly to our search through PT paths (Algorithm 2), MNTDP-D evaluates only $L + 1$ paths per problem, however, the search space does not include novel combinations of pre-trained modules which prevents the method from achieving few-shot transfer, in addition to being unable to achieve latent transfer. In contrast, our approach can achieve all three types of forward transfer.

While LMC (Ostapenko et al., 2021) also approximates each module's input distribution, PICLE's PT uses orders of magnitude fewer parameters and unites the approximations within a probabilistic model, allowing us to define a prior over the choice of modules and incorporate additional assumptions in a principled way. For each layer, LMC computes a linear combination of the outputs of all available pre-trained modules, thus, the trained model grows linearly with the number of solved problems, preventing LMC from scaling well. Finally, LMC's performance deteriorates when applied to long problem sequences. See Appendix C for comparison to works in the field of AutoML.

**Reproducibility Statement.** We have taken multiple steps to ensure the reproducibility of the presented results. With respect to our claims, we provide derivation of the posterior distribution used for PT paths (Eq. 2) in Appendix D. Moreover, we concretely specify the prior which we use for PT paths in Appendix E, along with the derivation we used to select its hyperparameter.

With respect to our experiments, we detail our experimental setup, along with all hyperparameters used for both our algorithm and the baselines in Appendix G. The compositional benchmarks and the neural architecture used for them are described in Appendix H. Similarly, the CTrL benchmarks and the neural architecture used for them are described in Appendix K.

Finally, PICLE's source code is available at `https://github.com/LazarValkov/PICLE`.

**Acknowledgements.** This research was partially supported by The NSF National AI Institute for Foundations of Machine Learning (IFML), ARO award #W911NF-21-1-0009, and DARPA award #HR00112320018.

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

## A  LIMITATIONS

Our method's limitations reveal directions for future work. First, PICLE does not search over all possible paths. e.g., paths which transfer both the initial and final layers, and learning new modules in between, could accomplish perceptual and latent transfer simultaneously. This can be readily addressed within our framework by employing a suitable probabilistic model. Second, the prior used when searching over PT paths relies on accuracy, making it difficult to be applied to problem sequences with different tasks, e.g. classification and regression. Third, the prior we used when searching over NT paths precludes us from finding novel pre-trained suffixes, which could be addressed by assigning non-zero probability to suffixes which are not derived from previous solutions.

## B  EXPERIMENTAL MEASUREMENTS

We use the following three measurements to assess the performance of each continual learning (CL) algorithm. First, we compute the average accuracy $\mathcal{A}$ across all problems after the last problem is solved. An algorithm's average accuracy after it is trained on some problem sequence $S = (\Psi_1, \Psi_2, ..., \Psi_{|S|})$ is computed as:

$$\mathcal{A}(S) = \frac{1}{|S|} \sum_{t=1}^{|S|} \mathcal{A}^{|S|}(\Psi_t) \tag{4}$$

where $\Psi_t$ denotes the $t$-th problem and $\mathcal{A}^{|S|}(\Psi_t)$ denotes a model's accuracy on this problem after all problems in the sequence have been solved. Second, we compute the forward transfer on the last problem only, $Tr^{-1}$, for sequences in which the performance on the last problem diagnoses an CL property. We compute it as the difference between the final accuracy on the last problem by a CL algorithm, and the accuracy of a standalone baseline, $\mathcal{A}_{\text{SA}}$:

$$Tr^{-1}(S) = \mathcal{A}^{|S|}(\Psi_{|S|}) - \mathcal{A}_{\text{SA}}(\Psi_{|S|}). \tag{5}$$

Finally, we compute the average forgetting experienced by a CL algorithm, as the average difference between the accuracy it achieves on a problem at the end of the sequence, and the accuracy it achieved on this problem initially:

$$\mathcal{F}(S) = \frac{1}{|S|} \sum_{t=1}^{|S|} \mathcal{A}^{|S|}(\Psi_t) - \mathcal{A}^t(\Psi_t) \tag{6}$$

## C  RELATED WORK - AUTOML

The work presented in this paper automates the process of choosing the best path. As such, the setting is similar to that of AutoML (He et al., 2021) in which the algorithms aim to automate different aspects of the process of applying machine learning to a problem. In particular, our setting is similar to hyperparameter optimisation (HPO) (Yu & Zhu, 2020) and neural architecture search (NAS) (Elsken et al., 2019). HPO aims to optimise different hyperparameters related to the training procedure (e.g. learning rate, choice of optimizer, magnitude of regularisation) and the neural architecture (e.g. number of hidden layers, number of hidden units, choice of activation functions). Categorical hyperpamaters are typically encoded as a one-of-k embedding. However, in our setting, embedding all of the selected modules would result in an embedding of size $\mathcal{O}(t^{l_{\min}})$ which would make the sample complexity of our surrogate model prohibitively high. NAS is a sub-setting of HPO which specialises to searching through more elaborate neural architectures. For this purpose, NAS methods also study how to best featurize a neural architecture. However, the approaches expressed in literature are specific to the respective neural architectures and aren't applicable to our setting. Despite the differences, there are also similarities between previous work in HPO and NAS, and our approach. For instance, most HPO and NAS approaches make use of a surrogate function. Moreover, it is common to use Bayesian optimisation with a Gaussian processes (GP) as a

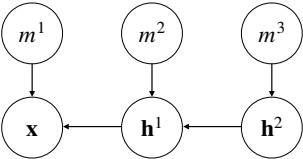

Figure 4: Our probabilistic model for a PT path with three pre-trained modules, $m^1, m^2, m^3$ and their respective inputs $\mathbf{x}, \mathbf{h}^1$ and $\mathbf{h}^2$.

surrogate function in HPO. Kandasamy et al. (2018) use Bayesian optimisation with a GP for NAS. Similarly to our work, they define their own kernel function to compute the similarity between two neural architectures. The idea of early stopping with Bayesian optimisation has also been explored before. For this purpose, Nguyen et al. (2017) use EI, Lorenz et al. (2015) use the probability of improvement (PI) and Makarova et al. (2022) use a measure based on the lower confidence bound (LCB). Makarova et al. (2022) compare all 3 approaches and show that while using EI and PI has a higher chance of stopping before finding an optimal solution, they also evaluate significantly fewer different configurations.

## D  PT PATHS - POSTERIOR DERIVATION

In this section we show how we approximate the posterior distribution $p(m^1, ..., m^l|\mathbf{x}_1, ..., \mathbf{x}_N)$. To ease the presentation, and without loss of generality, we set the number of pre-trained modules $l = 3$. The graphical model which captures the joint distribution is presented Figure 4. Next, we express the joint distribution in terms of quantities which we can approximate.

$$p(m^1, m^2, m^3, \mathbf{x}, \mathbf{h}^1, \mathbf{h}^2) = p(m^1)p(m^2)p(m^3)p(\mathbf{x}|\mathbf{h}^1, m^1)p(\mathbf{h}^1|\mathbf{h}^2, m^2)p(\mathbf{h}^2|m^3)$$

Here, $p(\mathbf{x}|\mathbf{h}^1, m^1)$ can be expressed as:

$$p(\mathbf{x}|\mathbf{h}^1, m^1) = \frac{p(\mathbf{x}, \mathbf{h}^1, m^1)}{p(\mathbf{h}^1, m^1)} = \frac{p(\mathbf{h}^1|\mathbf{x}, m^1)p(\mathbf{x}|m^1)p(m^1)}{p(\mathbf{h}^1)p(m^1)} = \frac{p(\mathbf{h}^1|\mathbf{x}, m^1)p(\mathbf{x}|m^1)}{\sum_{m^{2'}} p(\mathbf{h}^1|m^{2'})p(m^{2'})}.$$

Moreover, $p(\mathbf{h}^1|\mathbf{h}^2, m^2)$ can be expressed as:

$$p(\mathbf{h}^1|\mathbf{h}^2, m^2) = \frac{p(\mathbf{h}^1, \mathbf{h}^2, m^2)}{p(\mathbf{h}^2, m^2)} = \frac{p(\mathbf{h}^2|\mathbf{h}^1, m^2)p(\mathbf{h}^1|m^2)p(m^2)}{p(\mathbf{h}^2)p(m^2)} = \frac{p(\mathbf{h}^2|\mathbf{h}^1, m^2)p(\mathbf{h}^1|m^2)}{\sum_{m^{3'}} p(\mathbf{h}^2|m^{3'})p(m^{3'})}.$$

Therefore, the joint distribution can be expressed as:

$$\begin{aligned} &p(m^1, m^2, m^3, \mathbf{x}_1, \mathbf{h}^1, \mathbf{h}^2) \\ &= p(m^1)p(m^2)p(m^3) \frac{p(\mathbf{h}^1|\mathbf{x}, m^1)p(\mathbf{x}|m^1)}{\sum_{m^{2'}} p(\mathbf{h}^1|m^{2'})p(m^{2'})} \frac{p(\mathbf{h}^2|\mathbf{h}^1, m^2)p(\mathbf{h}^1|m^2)}{\sum_{m^{3'}} p(\mathbf{h}^2|m^{3'})p(m^{3'})} p(\mathbf{h}^2|m^3). \end{aligned}$$

(7)

Here, $p(\mathbf{h}^1|\mathbf{x}, m^1)$ and $p(\mathbf{h}^2|\mathbf{h}^1, m^2)$ define a distribution over the values of a hidden activation, given the module which produced it and said module's input. However, this hidden activation value is given by a deterministic transformation, $\mathbf{h}^i = m^i(\mathbf{h}^{i-1})$. Therefore, we can model them using the Dirac delta function, $\delta$: $p(\mathbf{h}^i|\mathbf{h}^{i-1}, m^i) = \delta(\mathbf{h}^i - m^i(\mathbf{h}^{i-1}))$. This function has the property

that $\int_{-\infty}^{\infty} f(z)\delta(z-c)dz = f(c)$, which we use next in order to simplify the posterior. We write:

$$p(m^1, m^2, m^3|\mathbf{x}) \propto p(m^1, m^2, m^3, \mathbf{x})$$

$$= \int\int p(m^1, m^2, m^3, \mathbf{x}, \mathbf{h}^{1'}, \mathbf{h}^{2'})d\mathbf{h}^{1'}d\mathbf{h}^{2'}$$

$$= \int\int p(m^1)p(m^2)p(m^3)\frac{p(\mathbf{h}^{1'}|\mathbf{x}, m^1)p(\mathbf{x}|m^1)}{\sum_{m^{2'}} p(\mathbf{h}^{1'}|m^{2'})p(m^{2'})}\frac{p(\mathbf{h}^{2'}|\mathbf{h}^{1'}, m^2)p(\mathbf{h}^{1'}|m^2)}{\sum_{m^{3'}} p(\mathbf{h}^{2'}|m^{3'})p(m^{3'})}p(\mathbf{h}^{2'}|m^3)d\mathbf{h}^{1'}d\mathbf{h}^{2'}$$

$$= \int p(m^1)p(m^2)p(m^3)\frac{p(\mathbf{x}|m^1)}{\sum_{m^{2'}} p(\mathbf{h}^1|m^{2'})p(m^{2'})}\frac{p(\mathbf{h}^{2'}|\mathbf{h}^1, m^2)p(\mathbf{h}^1|m^2)}{\sum_{m^{3'}} p(\mathbf{h}^{2'}|m^{3'})p(m^{3'})}p(\mathbf{h}^{2'}|m^3)d\mathbf{h}^{2'}$$

$$= p(m^1)p(m^2)p(m^3)\frac{p(\mathbf{x}|m^1)}{\sum_{m^{2'}} p(\mathbf{h}^1|m^{2'})p(m^{2'})}\frac{p(\mathbf{h}^1|m^2)}{\sum_{m^{3'}} p(\mathbf{h}^2|m^{3'})p(m^{3'})}p(\mathbf{h}^2|m^3)$$

$$\tag{8}$$

where $\mathbf{h}^1$ and $\mathbf{h}^2$ are the hidden activations obtained by processing the given $\mathbf{x}$ with the selected modules $m^1$ and $m^2$.

## E  DEFINING THE PRIOR FOR PT PATHS

Computing Eq. 2 requires us to define a prior distribution over the choice of a pre-trained module, $p(m^i)$. Assume that two modules $m_a^i$ and $m_b^i$ are trained using two different paths on two different problems. Also assume that the model trained on problem $\Psi_a$ achieved $\mathcal{A}^a(\Psi_a)$ validation accuracy after training, while the model trained on problem $\Psi_b$ achieved a higher validation accuracy $\mathcal{A}^b(\Psi_b) = \mathcal{A}^a(\Psi_a) + \Delta$, for $\Delta > 0$. We hypothesise that the module, whose model achieved the higher accuracy after training, is likely to compute a transformation of its input which is more likely to be useful for other problems. Therefore, if $m_a^i$ and $m_b^i$ have a similar likelihood for a given set of training data points, we would like to give preference to using $m_b^i$. To this end, we define the prior distribution in terms of a module's original accuracy using the *softmax* function as follows:

$$p(m_j^i) = \frac{\exp\{\mathcal{A}^j(\Psi_j)/T\}}{\sum_{m_{j'}^i \in \mathcal{L}^i} \exp\{\mathcal{A}^{j'}(\Psi_{j'})/T\}}. \tag{9}$$

Here $T$ is the temperature hyperparameter which we compute as follows. Suppose that, for a given set of input $\mathbf{x}$, we have selected the first $i-1$ modules and have computed the input to the $i$th module $\mathbf{h}^{i-1}$. Moreover, suppose that the likelihood of module $m_a^i$ is $\xi$ times higher than the likelihood of $m_b^i$, i.e. that $p(\mathbf{h}^{i-1}|m_a^i) = \xi p(\mathbf{h}^{i-1}|m_b^i)$. However, because the model of $m_b^i$ was trained to a higher accuracy, we would like to give equal preference to $m_a^i$ and $m_b^i$. Therefore, we would like to set the hyperparameter $T$ so that the posterior of the path using $m_a^i$ and the posterior of the path using $m_b^i$ are equal. Using Eq. 2 we can express this as:

$$p(m^1, ..., m_a^i|\mathbf{x}) = p(m^1, ..., m_b^i|\mathbf{x})$$

$$p(m_a^i)p(\mathbf{h}^{i-1}|m_a^i) = p(m_b^i)p(\mathbf{h}^{i-1}|m_b^i)$$

$$\frac{p(m_a^i)}{p(m_b^i)} = \frac{p(\mathbf{h}^{i-1}|m_b^i)}{p(\mathbf{h}^{i-1}|m_a^i)}$$

$$\frac{\exp\{\mathcal{A}^a(\Psi_a)/T\}}{\exp\{(\mathcal{A}^a(\Psi_a) + \Delta)/T\}} = \frac{p(\mathbf{h}^{i-1}|m_b^i)}{p(\mathbf{h}^{i-1}|m_a^i)} \tag{10}$$

$$\frac{\mathcal{A}^a(\Psi_a)}{T} - \frac{\mathcal{A}^a(\Psi_a) + \Delta}{T} = \log p(\mathbf{h}^{i-1}|m_b^i) - \log p(\mathbf{h}^{i-1}|m_a^i)$$

$$T = \frac{-\Delta}{\log p(\mathbf{h}^{i-1}|m_b^i) - \log p(\mathbf{h}^{i-1}|m_a^i)}$$

$$T = \frac{\Delta}{\xi}.$$

We can then use Eq. 10 in order to determine the value of T. To do this, we need to decide how much difference in log likelihood should an advantage in accuracy compensate for.

## F    FUNCTION DISTANCE FOR NT PATHS

The Gaussian Process which we defined in Section 5 relies on a distance between two functions. For this purpose, we make use of the standard Euclidean distance in function space. Let the inner product between $f : \Omega \to \mathbb{R}^r$ and $g : \Omega \to \mathbb{R}^r$ be $\langle f , g \rangle = \int_\Omega f(\mathbf{z}) \cdot g(\mathbf{z}) d\mathbf{z}$ . This allows us to define the distance between two functions as:

$$d(f, g) := ||f - g|| = \sqrt{\langle f - g , f - g \rangle} = \sqrt{\int_\Omega (f(\mathbf{z}) - g(\mathbf{z})) \cdot (f(\mathbf{z}) - g(\mathbf{z})) d\mathbf{z}} \ .$$

## G    EXPERIMENTAL SETUP

In our experiments, we make the training process deterministic, so that the difference in performance can be accredited only to the LML algorithm, and not due to randomness introduced during training. For this purpose, we fix the random initialisation of new parameters to be problem and path-specific. In other words, for a given problem, if a model with the same path is instantiated twice, it will have the same initial values for its new randomly initialised parameters. Moreover, we fix the sequence of randomly selected mini batches seen during training to be the same for a given problem. Finally, as we use PyTorch for our experiments, we fix the random seed and use the command "torch.use_deterministic_algorithms(True)". The overall results is that, for a given problem and a given library, evaluating the same path will always result in the same performance, even across different modular LML algorithms.

All experiments are implemented using PyTorch 1.11.0  (Paszke et al., 2019). We also use GPy's (GPy, since 2012) implementation of a Gaussian process. We run each LML algorithm on a single sequence, on a separate GPU. All experiments are run on a single machine with two Tesla P100 GPUs with 16 GB VRAM, 64-core CPU of the following model: "Intel(R) Xeon(R) Gold 5218 CPU @ 2.30GHz", and 377 GB RAM.

### G.1    ALGORITHM HYPERPARAMETERS

We now provide the implementation details and hyperparameters for each of the CL algorithms, which we evaluate on the benchmark suite with compositional tasks (BELL) and the CTrL benchmark suite.

**PICLE**    When searching through PT paths, we use a prior with softmax temperature ($T = 0.001$) for BELL and ($T = 0.6247744509446062$) for CTrL. When approximating a module's input distribution, we project its inputs to $k = 20$ dimensions. When searching through NT paths, we use GPy's GPy (since 2012) GP implementation. We combine its prediction using UCB Srinivas et al. (2009) with $\beta = 2$. We start from $l_{min} = 3$ and at the end of the problem store 40 of the training inputs to the $(L-2)$th layer. We set the number of paths evaluated during the Bayesian optimisation portion of our NT search, to $c = L + l_{min}$, which is constant in the number of solved problems.

**MNTDP-D Veniat et al. (2020)**    When selecting the closest previous solution, we use the 5-nearest-neighbours and the KNN classifier provided by *sklearn* Pedregosa et al. (2011).

**LMC Ostapenko et al. (2021)**    We use the implementation provided by the authors [2]. We run the task-aware version of the algorithm with otherwise the hyperparameters provided by the authors for the CTrL sequence.

**HOUDINI Valkov et al. (2018)**    We fixed the choice of modular neural architecture and use exhaustive search through the set of all paths. We evaluate $2L + t$ where $t$ is the number of solved problems, letting its computational requirements to scale linearly with the number of solved problems.

---

[2]Accessed at https://github.com/oleksost/LMC

**RS**    Random search randomly selects paths from the set of all paths for evaluation. However, many of the selected paths can consist only of pre-trained modules, which are cheap to evaluate since we don't need to train new parameters. To keep the results comparable, we instead limit the amount of time which can be taken by RS to solve a single problem to $(2L + t)\tau_{\text{SA}}$ where $\tau_{\text{SA}}$ is the amount of time it takes to train a randomly initialised network.

**O-EWC (Schwarz et al., 2018; Chaudhry et al., 2018)**    We use the implementation provided by CL-Gym Mirzadeh & Ghasemzadeh (2021)[3], with $\lambda = 1000$ (following Ostapenko et al. (2021)) and 256 samples to approximate the diagonal of the Fisher information matrix.

**ER (Chaudhry et al., 2019)**    For experience replay, we use a Ring Buffer with 15 examples per class (following Veniat et al. (2020)) for the multi-class classification tasks in CTrL and 40 for the binary classification compositional tasks in BELL. In this way, ER stores at least as many example per problem as PICLE.

## H    BELL - BENCHMARKS FOR LIFELONG LEARNING

We identify the following CL desiderata Valkov et al. (2018); Veniat et al. (2020):

1. *Stability* - The CL algorithm should be able to "remember" previous problems. As the algorithm solves new problems, its generalisation performance on previous problems should not drop drastically, i.e. no *catastrophic forgetting*.

2. *Plasticity* - The CL algorithm should be able to solve new problems. The algorithm's performance on a new problem should not be worse than that of a standalone baseline.

3. *Forward transfer* - The CL algorithm should be able to reuse previously obtained knowledge. As a result, its performance on a new problem should be greater than that of a standalone baseline, whenever possible.

   (a) *Perceptual Transfer* - The CL algorithm should be able to transfer knowledge across problems with similar input domains.

   (b) *Latent Transfer* - The CL algorithm should be able to transfer knowledge across problems with dissimilar input domains, including disparate input distributions or different input spaces.

   (c) *Few-shot Transfer* - The CL algorithm should be able to solve new problems, represented only by a few training data points, if the new problem can be solved using the already accumulated knowledge.

4. *Backward Transfer* - The CL algorithm should be able to use its newly obtained knowledge to improve its performance on previous problems. As a result, its performance on a previous problem should be greater after learning a new problem, than the initial performance on the previous problem, whenever possible.

5. *Scalability* - The CL algorithm should be applicable to a large number of problems. Therefore, the memory (RAM) and computational requirements should scale sub-linearly with the number of problems.

We now introduce *BELL* - a suite of benchmarks for evaluating the aforementioned CL properties. We assume compositional tasks and then generate various continual learning sequences, each of which evaluates one or two of the desired properties. Running a CL algorithm on all sequences then allows us to asses which properties are present and which are missing. This builds upon the CTrL benchmark suite  (Veniat et al., 2020), which defines different sequences of image classification tasks, namely $S^{\text{pl}}$, $S^{-}$, $S^{\text{out}}$, $S^{\text{in}}$, $S^{+}$ and $S^{\text{long}}$. They evaluate plasticity, perceptual transfer, latent transfer, catastrophic forgetting, backward transfer and scalability. We define these sequences similarly but for problems with compositional tasks. This allows us to introduce new sequences which evaluate new CL properties ($S^{sp}$ and $S^{few}$). We also introduce new more challenging sequences ($S^{out*}$ and $S^{out**}$).

---

[3] Accessed at https://github.com/imirzadeh/CL-Gym

| S | Sequence Pattern | CL |
|---|---|---|
| $S^{pl}$ | $[\Psi_1^+, \Psi_2^+, \Psi_3^+, \Psi_4^+, \Psi_5^+, \Psi_6^+]$ | 1., 2. |
| $S^-$ | $[\Psi_1^+, \Psi_2^-, \Psi_3^-, \Psi_4^-, \Psi_5^-, \Psi_1^-]$ | 3. |
| $S^{out}$ | $[\Psi_1^+, \Psi_2^-, \Psi_3^-, \Psi_4^-, \Psi_5^-, \Psi_6^- = (D_1, h_1, g_6)]$ | 3.a |
| $S^{out*}$ | $[\Psi_1^-, \Psi_2^+ = (D_1, h_1, g_2), \Psi_3^-, \Psi_4^-, \Psi_5^-, \Psi_6^- = (D_1, h_1, g_6)]$ | 3.a |
| $S^{out**}$ | $[\Psi_1^-, \Psi_2^+ = (D_1, h_1, g_2), \Psi_3^-, \Psi_4^-, \Psi_5^-, \Psi_1^-]$ | 3.a |
| $S^{in}$ | $[\Psi_1^+, \Psi_2^-, \Psi_3^-, \Psi_4^-, \Psi_5^-, \Psi_6^- = (D_6, h_6, g_1)]$ | 3.b |
| $S^{sp}$ | $[\Psi_1^+, \Psi_2^-, \Psi_3^-, \Psi_4^-, \Psi_5^-, \Psi_6^- = (D_6, h_6, g_1)]$ | 3.b |
| $S^{few}$ | $[\Psi_1^+, \Psi_2^+, \Psi_3^-, \Psi_4^- = (D_1, h_1, g_4), \Psi_5^-, \Psi_6^{--} = (D_2, h_2, g_4)]$ | 3.c |
| $S^+$ | $[\Psi_1^-, \Psi_2^-, \Psi_3^-, \Psi_4^-, \Psi_5^-, \Psi_1^+]$ | 4. |
| $S^{long}$ | $[\Psi_i]_{i=1}^{60}$ | 5. |

Table 5: A list of all of the different CL sequences in *BELL*, each of which evaluates different CL properties. The first column contains the sequence's name, the second shows the sequence's pattern and the third column lists the CL properties evaluated by this sequence.

We assume compositional tasks and represent each problem as a triple $\Psi_t = (D_j, h_j, g_k)$ where $D_j$ is the distribution of the inputs and $h_j$ and $g_k$ constitute the labelling function, i.e. $g \circ h$ is used to label each input. We refer to $h_j$ as the lower labelling sub-function and to $g_k$ as the upper labelling sub-function. We use the indices $j$ and $k$ to indicate whether the corresponding labelling sub-function has occurred before in the sequence ($j < t, k < t$) or if it is new and randomly selected ($j = t, k = t$). For brevity, if $j = t$ and $k = t$ which means that both labelling sub-functions are new, we don't write out the whole triple but only $\Psi_t$. By repeating previously labelling sub-functions, we can control what knowledge can be transferred in each of the define sequences. In turn, this allows us to evaluate different CL properties. We use $\Psi^+$ to indicate that the dataset generated for this problem is sufficient to learn a well generalising approximation without transferring knowledge. On the other hand, $\Psi^-$ indicates that the CL algorithm cannot achieve good generalisation on this problem without transferring knowledge. Finally, $\Psi^{--}$ indicates that the generated training dataset consists of only a few datapoints, e.g. 10.

A complete list of the different sequences in *BELL* is presented in Table 5. Following Veniat et al. (2020), we set the sequence length of most sequences to 6 which, as we show in the experiments section, is sufficient for evaluating different CL properties. Next, we separately present each sequence, detailing which CL properties it evaluates.

**Plasticity** and **Stability**: The sequence $S^{pl} = [\Psi_1^+, \Psi_2^+, \Psi_3^+, \Psi_4^+, \Psi_5^+, \Psi_6^+]$ consists of 6 distinct problems, each of which has a different input domain and a different task. Moreover, each of the generated datasets has a sufficient number of data points as not to necessitate transfer. Therefore, this sequence evaluates a CL algorithm's ability to learn distinct problems, i.e. its plasticity (1.). Moreover, this sequence can be used to evaluate an algorithm's stability (2.) by assessing its performance after training on all problems and checking for forgetting.

**Forward Transfer**: Most of our sequences are dedicated to evaluate different types of forward transfer. To begin with, in the sequence $S^- = [\Psi_1^+, \Psi_2^-, \Psi_3^-, \Psi_4^-, \Psi_5^-, \Psi_1^-]$ the first and the last datasets represent the same problem, however, the last dataset has fewer data points. Therefore, a CL algorithm would need to transfer the knowledge acquired from solving the first problem, thus, demonstrating its ability to perform overall forward transfer (3.).

**Perceptual Forward Transfer**: We introduce three different sequences for evaluating perceptual transfer (3.a). First, in $S^{out} = [\Psi_1^+, \Psi_2^-, \Psi_3^-, \Psi_4^-, \Psi_5^-, \Psi_6^- = (D_1, h_1, g_6)]$ the last problem has the same input domain and input-processing target function $h_1$ as in problem 1. However, the last problem's dataset is small, therefore, a CL algorithm needs to perform perceptual transfer from the first problem, which is described by a large dataset. Second, $S^{out*} = [\Psi_1^-, \Psi_2^+ = (D_1, h_1, g_2), \Psi_3^-, \Psi_4^-, \Psi_5^-, \Psi_6^- = (D_1, h_1, g_6)]$ shares the same input distributions and lower labelling sub-function $h_1$ across problems $\Psi_1$, $\Psi_2$ and $\Psi_6$. Therefore, a CL algorithm needs to decide whether to transfer knowledge obtained from the first or from the second problem. Third, the sequence $S^{out**} = [\Psi_1^-, \Psi_2^+ = (D_1, h_1, g_2), \Psi_3^-, \Psi_4^-, \Psi_5^-, \Psi_1^-]$ is similar to the preceding one, with the distinction that the last problem is the same as the first. In this sequence, a CL algorithm needs to decide between reusing knowledge acquired from solving the same problem ($\Psi_1$), or to transfer perceptual knowledge from a more different problem ($\Psi_2$). Overall, these three sequences are de-

signed to be increasingly more challenging in order to distinguish between different CL algorithms which are capable of perceptual transfer to a different extent.

**Latent Forward Transfer**: Currently, we define two sequences to assess an algorithm's ability to transfer latent knowledge. Firstly, in $S^{in} = [\Psi_1^+, \Psi_2^-, \Psi_3^-, \Psi_4^-, \Psi_5^-, \Psi_6^- = (D_6, h_6, g_1)]$ the last problem has the same upper labelling sub-function as the first problem. However, the two problems' input distributions and lower labelling sub-functions are different. Therefore, a CL algorithm would need to transfer knowledge across different input domains. Secondly, the sequence $S^{sp} = [\Psi_1^+, \Psi_2^-, \Psi_3^-, \Psi_4^-, \Psi_5^-, \Psi_6^- = (D_6, h_6, g_1)]$ is simiarly defined, however, the input distribution of the last problem is also defined on a different input space from the input space of the first problem. Therefore, an algorithm would need to transfer knowledge across different input spaces.

**Few-shot Forward Transfer:** In order to evaluate this property, we introduce the following sequence, in which the first two problems are different from the rest of the sequences: $S^{few} = [\Psi_1^+ = (D_1, h_1), \Psi_2^+ = (D_2, h_2), \Psi_3^-, \Psi_4^- = (D_1, h_1, g_4), \Psi_5^-, \Psi_6^{--} = (D_2, h_2, g_4)]$. The labelling functions of the first two problems are simpler, each consisting only of a lower labelling sub-function. This is done in order to provide a CL algorithm with more supervision on how to approximate $h_1$ and $g_1$ more accurately. The fourth problem $\Psi_4$ in this sequence then shares the same input domain and lower lableling sub-function as the first problem, but introduces a new upper labelling sub-function $g_4$. The last problem then shares the input domain and the lower labelling sub-function of $\Psi_2$, while also sharing the upper labelling sub-function of problem $\Psi_4$. Moreover, the last problem's training dataset consists of only a few data points. Therefore, a CL algorithm would need to reuse its approximations of $h_2$ and $g_4$ in a novel manner in order to solve the last problem.

**Backward Transfer:** The sequence $S^+ = [\Psi_1^-, \Psi_2^-, \Psi_3^-, \Psi_4^-, \Psi_5^-, \Psi_1^+]$ has the same first and last problem. However, the first dataset has significantly less data points than the last. Ideally, a CL algorithm should use the knowledge acquired after solving the last problem in order to improve its performance on the first problem. While this sequence represents a starting point for evaluating backward transfer, it is possible to introduce other sequences, representing more elaborate evaluations. For instance, introducing sequences which evaluate perceptual and latent backward transfer separately. However, as backward transfer is not the focus of this paper, this is left for future work.

**Scalability:** This property can be evaluated using a long sequence of problems. For this purpose we define $S^{long} = [\Psi_i]_{i=1}^{60}$ which consists of 60 problems, each randomly selected with replacement from a set of problems. Most problems are represented by a small dataset, $\Psi_t^-$. Each of the first 50 problems has a $\frac{1}{3}$ probability of being represented by a large dataset, $\Psi_t^+$. Each problem also has a $\frac{1}{10}$ probability of being represented by an extra small dataset, $\Psi_t^{--}$.

Next, we present a set of problems which can be used together with the aforementioned sequence definitions in order to evaluate CL algorithms.

### H.1 COMPOSITIONAL PROBLEMS

To implement the sequences defined above, one needs to define a set of compositional problems. To this end, we define 9 different pairs of an input domain and a lower labelling sub-function, $\{(D_i, h_i)\}_{i=1}^9$. Moreover, we define 16 different upper labelling sub-functions $\{g_i\}_{i=1}^{16}$. These can be combined into a total of 144 different compositional problems.

First, we define 9 image multi-class classification tasks, which all share input and output spaces $\mathbb{R}^{28 \times 28} \rightarrow \mathbb{R}^8$, but each have a different input distribution $D_i$ and a domain-specific labelling function $h_i$. Concretely, we start with the following image classification datasets: MNIST (Le-Cun et al., 2010), Fashion MNIST (FMNIST) (Xiao et al., 2017), EMNIST (Cohen et al., 2017) and Kuzushiji49 (KMNIST) (Clanuwat et al., 2018). Since some of the classes in KMNIST have significantly fewer training data points, we only use the 33 classes with the following indices: $[0, 1, 2, 4 : 12, 15, 17 : 21, 24 : 28, 30, 34, 35, 37 : 41, 46, 47]$, as they have a sufficient number of associated data points. We split the image datasets into smaller 8-class classification datasets. We use an index $i$ to denote the different splits of the same original dataset. For instance EMNIST$_2$ represents the third split of EMNIST, corresponding to a classification task among the letters form 'i' to 'p'. As another example, MNIST$_1$ represents the only split of MNIST, corresponding to a classification tasks among the digits from 0 to 7. Using this, we end up with the following 9 image datasets: MNIST$_1$, FMNIST$_1$, $\{$EMNIST$_i\}_{i=1}^3$, $\{$KMNIST$_i\}_{i=1}^4$. For each of these image datasets, we set aside 4800 validation images from the training dataset. We also keep the provided test images separate.

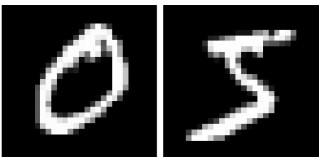

Figure 5: An illustration of the four two-dimensional patterns which are used by the four $g^{(2)}$ functions to label the input coordinates. Green indicates a positive label, and red indicates a negative label.

Second, we define a set of binary classification tasks, which map $\mathbb{R}^{16} \to \{0, 1\}$. Each task's labelling function $g_i$ receives two concatenated 8-dimensional one-hot encodings and returns a binary value, indicating if the given combination of 2 classes, represented by the input, fulfils a certain criteria. We further decompose the labelling function into $g_i(\mathbf{x}) = g_k^{(2)}(g_j^{(1)}(\mathbf{x}[:8]), g_j^{(1)}(\mathbf{x}[8:]))$.

Here, $g_j^{(1)}$ maps a one-hot encoding to an integer between 1 and 8. For instance, $g_1^{(1)}$ maps the first dimension to 1, the second to 2 and so on. As a result, we use $g^{(1)}$ to convert the initial input of two one-hot encodings to two-dimensional coordinates. We define 4 different $g^{(1)}$ mappings, where $g_1^{(1)}$ is defined as above, and $g_1^{(2)}, g_1^{(3)}$ and $g_1^{(4)}$ each map the dimensions to a different randomly selected integer between 1 and 8.

At the same time, each $g_k^{(2)} : \mathbb{R}^2 \to \{0, 1\}$ outputs whether a given two-dimensional coordinate is a part of a certain pattern or not. We define 4 different $g^{(2)}$ functions, each corresponding to one of 4 two-dimensional patterns, shown in Fig 5. In total, these functions need to label $8 * 8 = 64$ different two-dimensional coordinates.

We fuse the 4 different $g^{(1)}$ functions with the 4 different $g^{(2)}$ functions to define 16 different $g$ functions:

$$\{g_{(k-1)*4+j}(\mathbf{x}) = g_k^{(2)}(g_j^{(1)}(\mathbf{x}[:8]), g_j^{(1)}(\mathbf{x}[8:])), k \in \{1, 2, 3, 4\}, j \in \{1, 2, 3, 4\}\}.$$

Figure 6: An example input for $\Psi = (D_{\text{MNIST}_1}, h_{\text{MNIST}_1}, g = (g_{\text{XOR}}^{(2)}, g_1^{(1)}))$. These images are classified by $h_{\text{MNIST}}$ and then are mapped to the coordinates $(1, 6)$ by $g_1^{(1)}$ since they represent the first and sixth classes respectively. Afterwards, $g_{\text{XOR}}^{(2)}$ labels this input as 0, using the XOR pattern, shown in 5.

Finally, we can combine our 9 image classification datasets $\{(D_i, h_i)\}_{i=1}^9$ with our 16 binary classification tasks, in order to create 144 compositional problems $\{\Psi_{(k-1)*9+j} = (D_k, h_k, g_j), k \in \{1, ..., 9\}, j \in \{1, ..., 16\}\}$. The input to a problem $\Psi_i = (D_k, h_k, g_j)$ are two images sampled from $D_i$. Each image is labelled by $h_i$, each resulting in an eight-dimensional one-hot encoding of the corresponding image's class. The two one-hot encodings are then concatenated and labelled by $g_j$, which results in a binary label. An example for $\Psi = (D_{\text{MNIST}_1}, h_{\text{MNIST}_1}, g = (g_{\text{XOR}}^{(2)}, g_1^{(1)}))$ is shown in Fig 6.

Sequence $S^{\text{sp}}$ involves transferring across input spaces by having its last problem's input domain be defined over a different input space. To create this domain we flatten any randomly selected domain from $\mathbb{R}^{28 \times 28}$ to $\mathbb{R}^{784}$. This loses the images' spacial information and requires that a different neural architecture is applied to process those inputs.

## H.2 Realising the Sequences

To implement a sequence S of length $l$, we need to select $l$ concrete compositional problems which fit the pattern specified by said sequence. Let the sequence have $l^{(1)}$ different pairs of image domain and lower labelling sub-function, and $l^{(2)}$ different upper labelling sub-functions. For all sequences, apart from $S^{\text{long}}$, we select $l^{(1)}$ pairs of $(D_i, h_i)$ by sampling from the set of all possible image classification tasks, without replacement. Similarly, we select $l^{(2)}$ different upper labelling sub-functions by sampling without replacement from the set of available binary classification tasks $\{g_i\}_{i=1}^{16}$. For $S^{\text{long}}$, we use sampling with replacement.

If a problem's training dataset needs to be large, $\Psi_i^+$, we generate it according to the triple $n_{\text{tr}}^+ = (30000, \text{All}_{\text{tr}}, \text{All})$. The first value indicates that we generate 30000 data points in total. The second value indicates how many unique images from the ones set aside for training, are used when generating the inputs. In this case, we use all the available training images. The third value indicates how many out of the 64 unique two-dimensional coordinates, used by the upper labelling sub-function, are represented by the input images. In this case, we use all two-dimensional coordinates.

Some of the problems' training datasets are required to be small and to necessitate transfer. For sequences $S^-, S^{\text{out}}, S^{\text{out*}}, S^{\text{out**}}, S^{\text{few}}, S^+$, we generate the training datasets of each problem $\Psi^-$ using the triple $n_{\text{tr}}^- = (10000, 100, \text{All})$. This way, only 100 unique images are used to generate the training dataset, so solving the problem is likely to be difficult without perceptual transfer. The subset of unique images is randomly sampled and can be different between two problems which share an input domain. For sequences $S^{\text{in}}$ and $S^{\text{sp}}$, which evaluate latent transfer, we use the triple $n_{\text{tr}}^- = (10000, \text{All}_{\text{tr}}, 30)$. As a result, the generated datasets will only represent 30 out of 64 of the two-dimensional coordinates, which is not sufficient for learning the underlying two-dimensional pattern. Therefore, these problems will necessitate latent transfer. When generating a dataset for a problem $\Psi^-$ in the sequence $S^{\text{long}}$, we randomly choose between the two, namely between $(10000, 100, \text{All})$ and $(10000, \text{All}_{\text{tr}}, 30)$.

For the problems in which the training dataset needs to contain only a few data points, $\Psi^{--}$, we use the triple $n_{\text{tr}}^{--} = (10, 20, 10)$. This creates only 10 data points, representing 20 different images and 10 different two-dimensional patterns.

For problems with $\Psi^{--}$, we use the triple $n_{\text{val}}^{--} = (10, 20, 10)$ for generating the validation dataset. For the rest of the problems, we use the triple $n_{\text{val}}^{--} = (5000, \text{All}_{\text{val}}, \text{All})$. Finally, we generate all test datasets using the triple $n_{\text{test}}^{--} = (5000, \text{All}_{\text{test}}, \text{All})$.

## H.3 Neural Architecture and Training

Here, we present the neural architecture which we have found to be suitable for solving the aforementioned compositional problems.

We first define a convolutional neural network $\zeta_{\text{CNN}} : \mathbb{R}^{28 \times 28} \to \mathbb{R}^8$, suitable for processing images from the image classification datasets. We use a 5-layer architecture with *ReLU* hidden activations and a *softmax* output activation. The layers are as follows: *Conv2d(input_channels=1, output_channels=64, kernel_size=5, stride=2, padding=0), Conv2d(input_channels=64, output_channels=64, kernel_size=5, stride=2, padding=0), flatten, FC(4\*4\*64, 64), FC(64, 64), FC(64, 10)*. Here, *Conv2d* specified a two-dimensional convolutional layer and *FC* specifies a fully-connected layer.

Second, we define a fully-connected neural network for processing a concatenation of two 8-dimensional one-hot embeddings, $\zeta_{\text{MLP}} : \mathbb{R}^{16} \to \mathbb{R}^1$. It consists of 2 *FC* hidden layers with 64 hidden units and *RELU* hidden activations, followed by an output *FC* layer with a *sigmoid* activation.

For a compositional problem $\Psi_k = (D_i, h_i, g_j)$ the input is a 2-tuple of images, $(\mathbf{x}^1, \mathbf{x}^2)$ and the expected output is a binary classification. We solve it using the architecture $\zeta_{\text{comp}} = \zeta_{\text{MLP}}(concatenate(\zeta_{\text{CNN}}(\mathbf{x}^2), \zeta_{\text{CNN}}(\mathbf{x}^2)))$. This architecture processes each of the 2 input images with the same $\zeta_{\text{CNN}}$ model. Then the 2 outputs are concatenated and processed by a $\zeta_{\text{MLP}}$ model.

We represent this as a modular neural architecture by considering each of the 8 parameterised non-linear transformations to be a separate module. This increases the number of possible paths for each problem. As a result, for the 6th problem in a sequence, the number of possible paths is upper bounded by $\mathcal{O}(6^8 = 1679616)$. Therefore, in this setting, even sequences of length 6 are challenging for modular CL approaches.

The input space of the last problem of sequence $S^{\text{sp'}}$ is given by an 8-dimensional vector. Therefore, only for this problem, we replace $\zeta_{\text{CNN}}$ with a different architecture, $\zeta_{\text{FC}}$, which consists of two fully connected layers, with a hidden size of 64, and uses $ReLU$ as a hidden activation and *softmax* as its output activation.

We train new parameters to increase the log likelihood of the labels using the AdamW optimiser (Loshchilov & Hutter, 2017) with 0.00016 learning rate, and 0.97 weight decay. The training is done with a mini batch size of 32 and across 1200 epochs. We apply early stopping, based on the validation loss. We stop after 6000 updates without improvement and return the parameters which were logged to have had the best validation accuracy during training.

## I  ABLATION: GAUSSIAN APPROXIMATION FOR PT SEARCH

The probabilistic model which we use to search through PT paths (Eq. 1) relies on an approximation of the input distribution which a pre-trained module has been trained with. We proposed to first project samples from this distribution to $k$ dimensions using random projection, and then fit a multivariate Gaussian on the resulting samples. In this section we would like to evaluate three aspects of this approach. First, we would like to assess the usefulness of the resulting approximations for the purposes of selecting the correct input distribution. Second, we would like to assess the sensitivity of our approximations to the hyperparameter $k$. Third, we would like to compare our approach to Gaussian approximation of the original input space in order to determine whether we sacrifice performance.

To this end, we evaluate whether our approach is useful for distinguishing between a set of input distributions. We compare the approximations resulting from different choices of $k = \{10, 20, 40\}$. The resulting methods are referred to as $rp\_10$, $rp\_20$ and $rp\_40$ respectively. Moreover, we compare to the method of fitting a Gaussian on the original samples, without a random projection. Since this can lead to a singular covariance matrix, we make use of diagonal loading (Draper & Smith, 1998) in which we add a small constant ($10^{-8}$) to the diagonal of the computed sample covariance matrix in order to make it positive definite. We refer to the resulting method as *diag_loading*. We compare how well these approaches can distinguish between the 9 image datasets used in BELL. We chose to use the input images for our comparison since they have the highest dimension and their distributions should be the most difficult to approximate.

To evaluate one of the methods, we first use it to approximate all 9 input distributions using $N$ data points, resulting in 9 approximations, denoted as $\{q_i\}_i^9$. Second, for each input distribution $p_j$, we sample 100 different data points and use them to order the approximations in a descending order of their likelihood. Ideally, if the data points are sampled from the $j$-th distribution $p_j$, the corresponding approximation $q_j$ should have the highest likelihood, and thus should be the first in the list, i.e. should have an index equal to 0. We compute the index of $q_j$ in the ordered list and use it as an indication of how successfully the method has approximated $p_j$. We compute this index for each of the 9 distributions and report the average index, also referred to as the *average position*.

We evaluate each method for different choices of $N$, $N = \{50, 100, 500, 1000, 5000, 10000, 20000, 30000, 44000\}$. Moreover, we repeat all evaluations 5 times using different random seeds and report the mean and standard error of the average position. The results are reported in Fig. 7.

Our results show that directly modelling the original distribution with a Gaussian leads to sub-optimal performance. On the other hand, we observe that for $N \geq 500$, the methods which use random projection can always match the given data points with the correct distribution which they were sampled for. Surprisingly, for $N = 50$ and $N = 100$, diag_loading outperforms the other methods and can successfully identify the correct distribution of the given data points. Furthermore, we observe that for these values of $N$, decreasing the dimension $k$ that the data points are projected to leads to better performance of the methods that are based on random projection.

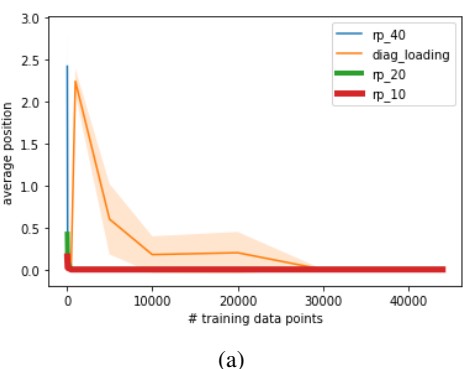
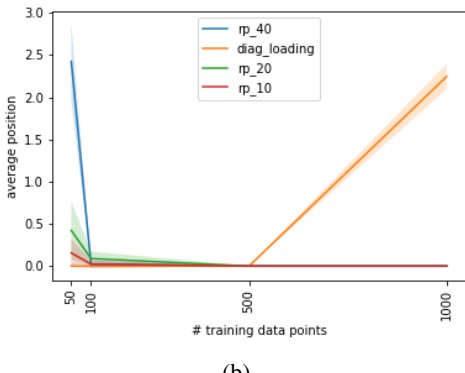

| (a) | (b) |

Figure 7: Comparison of different methods for approximating a module's input distribution. The x axis represents the number $N$ of data points used to compute an approximation. The y axis represents the average position, as defined in the main text, which indicates how well a method can approximate the distributions. The lower the average position is, the better the model performs. Figure a) presents a plot across all choices of $N$. Figure b) focuses on the first few values of $N$.

Overall, our results suggest that the approximations which we use for out probabilistic model for PT paths are effective when the new modules are trained on more than 100 data points. This seems like a reasonable requirement, as fewer points are likely to result in a sub-optimal performance.

## J    ABLATIONS: SEARCH THROUGH NT PATHS

Our search through NT paths (Algorithm 3) involves performing a Bayesian optimisation (BO) over NT paths with suffix-length $l_{min}$ (lines 4-8). In this section we evaluate different properties of our BO sub-routine. First, we assess its ability to accelerate the search for the optimal NT path. Second, we assess its early stopping capabilities. Third, we compare our kernel function to different alternatives.

For this purpose, we create a new sequence of compositional problems:

$$S^{\text{in}+} = [\Psi_1^+, \Psi_2^+, ..., \Psi_{15}^+, \Psi_{16}^- = (D_6, h_6, g_1)] \tag{11}$$

which involves all 16 upper labelling sub-functions $g$ of BELL. The last problem's dataset is generated according to the triple $n_{\text{tr}}^- = (10000, \text{All}_{\text{tr}}, 30)$, which states that only 30 out of the 64 possible two-dimensional patterns are represented in the dataset. As a result, non-perceptual transfer is necessary in order to maximise the performance on the final problem. We create 5 realisations of $S^{\text{in}+}$ with different randomly selected problems.

For each of the 5 realisations of the sequence $S^{\text{in}+}$, we run each competing method 10 times with 10 different random seeds. This results in $10 * 5 = 50$ evaluations per method which we average over when reporting its performance. For each method, we plot its maximum accuracy achieved per number of paths evaluated. We refer to our realisation of Bayesian optimisation as "BO_l2" as it uses the l2 norm to calculate a distance between functions (see Appendix F for details).

### J.1    SEARCH ACCELERATION

The first two paths which we select for evaluation are selected deterministically as follows. We calculate each suffix's distance to other suffixes, and choose the 2 suffixes whos average distance to others is the lowest. We do this because these 2 suffixes are likely to be more informative about the rest, allowing us to better approximate the rest's performance. In this section, we compare this decision to the alternative of randomly selecting the first two suffixes: referred to as "BO_l2_r". Moreover, we compare using Bayesian optimisation to just using random search, referred to as "RS". The results are shown in Fig. 8a. It can be observed that our BO implementation, BO_l2, greatly reduces the number of paths required to be evaluated before finding near-optimal performance. Note

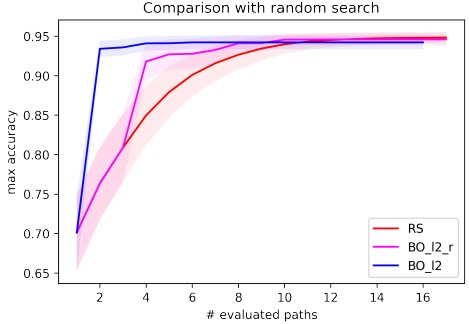 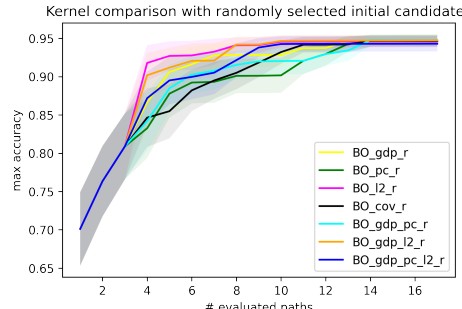

(a) Comparing to randomly selected initial points (BO_l2_rnd_init), and random search (RS).

(b) Comparing between different choices of GP kernels.

Figure 8: Comparing different design choices for our Bayesian optimisation algorithm $s_{\text{BO}}^{\text{NT},l}$.

that it does not eventually reach the very best performance due to early stopping (see next subsection). Overall, these results indicate that using Bayesian optimisation significantly accelerates the search, compared to random search.

## J.2 EARLY STOPPING

Using a Gaussian process (GP) makes it possible to detect when further improvement is unlikely, allowing us to perform early stopping. For this purpose, whenever we select a new pre-trained suffix during our Bayesian optimisation (line 5 of Algorithm 3), we can compute its Expected Improvement (EI): $\text{EI}(p_{\text{GP}}(f|\lambda, \boldsymbol{\lambda'}, \mathbf{f}))$. This tells us how much we expect the selected suffix's performance to improve the current best observed performance. If it is lower than a certain threshold, we can stop the Bayesian optimisation. EI-based early stopping has been previously suggested in Nguyen et al. (2017) and, similarly to Makarova et al. (2022), our preliminary experiments showed that it leads to fewer path evaluations, compared to using UCB for early stopping.

When evaluated on our ablation sequence $S^{\text{in}+}$, early stopping allowed our approach BO_l2 to evaluate 11.5 paths on average. In contrast, random search always had to evaluate all 17.

## J.3 ALTERNATIVE KERNELS

Next, we investigate different choices for a kernel function used by the Gaussian process for Bayesian optimisation. Our implementation BO_l2 uses the RBF kernel in order to convert a distance between functions into a similarity between functions. Instead, we can directly calculate different similarity measures $sim_v$ which can then be used with the following kernel:

$$k_v(\pi_j^{\text{NT},l}, \pi_k^{\text{NT},l}; Z) = \sigma_0^2 + \sigma_1^2 sim_v(\pi_j^{\text{NT},l}, \pi_k^{\text{NT},l}; Z) \tag{12}$$

where $\sigma_0$ and $\sigma_1$ are scalar hyperparameters that are optimised on the GP's training dataset, and $Z$ is a set of points from the functions' input space. We define the following similarity measurements for two scalar functions.

First, we can compute the covariance between the function's outputs which computes the linear relationship between the functions' outputs but also reflects the magnitude of the outputs. This leads to the following similarity:

$$sim_{\text{cov}}(f_1, f_2; Z) := COV(\cup_i f(z_i), \cup_i g(z_i)) . \tag{13}$$

Second, we can compute the sample Pearson correlation coefficient (Lee Rodgers & Nicewander, 1988), denoted as $PC$, between the functions' outputs. This captures the linear correlation of the functions' outputs while ignoring their magnitude. As a result, the similarity ranges between $[-1, 1]$. The similarity is defined as:

$$sim_{\text{pc}}(f_1, f_2; Z) := PC(\cup_i f(z_i), \cup_i g(z_i)) . \tag{14}$$

Third, we note that the aforementioned similarities are computed based on the functions' outputs which only reflect points in their output space. Instead, we can compare the functions' curvatures around each evaluation input $z_i$. For each input, we compute the gradient of each function's output with respect to its input and normalise it to have a unit norm. The dot product between the two resulting normalised gradients is then computed in order to capture the alignment between the two curvatures. This leads to the following similarity:

$$sim_{\text{gdp}}(f_1, f_2; Z) := \frac{1}{V} \sum_{i=1}^{V} \left( \frac{\nabla f_1(z_i)}{\|\nabla f_1(z_i)\|_2} \right) \cdot \left( \frac{\nabla f_2(z_i)}{\|\nabla f_2(z_i)\|_2} \right) \quad . \tag{15}$$

The 3 similarities defined above lead to 3 kernels, which in turn result in the following 3 BO algorithms: BO_cov_r, BO_pc_r, BO_gpd_r. Moreover, one can sum kernel functions in order to result in a new kernel function, which uses a combination of similarities. We combine different kernels which leads to the following algorithms: BO_gpd_pc_r, BO_gpd_l2_r and BO_gpd_l2_pc_r. For all algorithms, we use randomly selected initial paths, because the paths selected deterministically perform too well which makes it harder to compare algorithms. The resulting comparison is presented in Fig. 8b. Overall, BO_l2_r achieves the best anytime performance by finding well-performing paths more quickly. We also observe that combining the RBF kernel of BO_l2_r with other kernels does not result in an improvement.

## K  CTRL BENCHMARKS

The CTrL benchmark suite was introduced in Veniat et al. (2020). They define a number of sequences, based on seven image classfication tasks, namely: CIFAR10 and CIFAR100 (Krizhevsky et al., 2009), DTD (Cimpoi et al., 2014), SVHN (Netzer et al., 2011), MNIST (LeCun et al., 1998), RainbowMNIST (Finn et al., 2019), and Fashion MNIST (Xiao et al., 2017). All images are rescaled to 32x32 pixels in the RGB color format. CTrL was first to introduce the following sequences: $S^-$, $S^+$, $S^{\text{in}}$, $S^{\text{out}}$, $S^{\text{pl}}$ and $S^{\text{long}}$, which are defined similarly to our definitions. However, the difference is that they are defined for and implemented by image classification tasks. The last task in $S^{\text{in}}$, which evaluates non-perceptual transfer, is given by MNIST images with a different background color than the first task. The last task in $S^{\text{out}}$ is given by shuffling the output labels of the first task. $S^{\text{long}}$ has 100 tasks. For each task, they sample a random image dataset and a random subset of 5 classes to classify. The number of training data points is sampled according to a distribution that makes it more likely for later tasks to have small training datasets. In contrast to us, they use only 1 selection of tasks for each sequence, i.e. 1 realisation of each sequence. To generate the sequences, we use the code provided by the authors (Veniat & Ranzato, 2021).

Our experimental setup mirrors that used in (Ostapenko et al., 2021) in order for us to be able to compare to LMC using the official implementation. The modular neural architecture consists of 5 modules with a hidden size of 64. The last module is a linear transformation. The first 4 modules are identical and are a sequence of: 2-dimensional convolution (kernel size 3, stride 1, padding 2), batch normalisation (momentum 0.1), ReLU activation and 2-dimensional max-pool (kernel size 2, no stride and 0 padding).

A single network is trained with Adam Kingma & Ba (2014), with learning rate of $1e-3$ and weight decay of $1e-3$. The batch size is 64 and we train for 100 epochs. After each epoch, the current validation performance is evaluated. The best weights which resulted in the best validation performance, are restored at the end of the training.

### K.1  RESULTS WITH STANDARD DEVIATIONS

Table 6 shows our results on the short CTrL sequences, with included standard deviations.

## L  RESULTS - ORIGINAL CTRL SETUP

We also ran experiments on the original CTrL experimental setup (Veniat et al., 2020), which is outlined next. The neural architecture used is a small variant of ResNet18 architecture which is divided into 6 modules, each representing a different ResNet block (He et al., 2016). While the paper

| | | SA | O-EWC | ER | RS | HOUDINI | LMC | MNTDP-D | PICLE |
|---|---|---|---|---|---|---|---|---|---|
| $\mathcal{A}$ | $S^{\text{in}}$ | $65.4 \pm 0.4$ | $24.5 \pm 1.3$ | $46.4 \pm 1.2$ | $65.4 \pm 1.5$ | $64.6 \pm 0.9$ | $\mathbf{69.5 \pm 0.5}$ | $66.2 \pm 0.8$ | $68.3 \pm 0.4$ |
| | $S^{\text{out}}$ | $63.1 \pm 0.3$ | $46.9 \pm 10.7$ | $56.6 \pm 0.6$ | $63.8 \pm 1.$ | $66.5 \pm 0.5$ | $65.7 \pm 1.6$ | $\mathbf{66.8 \pm 0.6}$ | $\mathbf{66.8 \pm 0.6}$ |
| | $S^{\text{pl}}$ | $64. \pm 0.3$ | $43.6 \pm 2.9$ | $58.8 \pm 0.8$ | $63.7 \pm 0.6$ | $63.9 \pm 0.1$ | $62.3 \pm 2.$ | $64.6 \pm 0.5$ | $\mathbf{64.7 \pm 0.4}$ |
| | $S^{-}$ | $63 \pm 0.2$ | $48.3 \pm 9.2$ | $54.9 \pm 0.8$ | $64.0 \pm 0.4$ | $67.2 \pm 0.9$ | $65.8 \pm 1.8$ | $\mathbf{67.2 \pm 0.2}$ | $\mathbf{67.2 \pm 0.2}$ |
| | $S^{+}$ | $63.0 \pm 0.4$ | $45.9 \pm 4.5$ | $56.8 \pm 0.6$ | $63.2 \pm 0.4$ | $62.8 \pm 0.3$ | $59.7 \pm 0.8$ | $\mathbf{63.1 \pm 0.1}$ | $63. \pm 0.3$ |
| | **Avg.** | $63.7 \pm 1.1$ | $41.9 \pm 9.9$ | $54.7 \pm 4.8$ | $64.0 \pm 0.8$ | $65. \pm 1.8$ | $64.6 \pm 3.7$ | $65.6 \pm 1.7$ | $\mathbf{66. \pm 2.2}$ |
| $\mathcal{F}$ | **Avg.** | $0. \pm 0$ | $-20.6 \pm 12.0$ | $-4.8 \pm 2.0$ | $0. \pm 0$ | $0. \pm 0$ | $-0.7 \pm 0.8$ | $0. \pm 0$ | $\mathbf{0. \pm 0}$ |
| $Tr^{-1}$ | $S^{\text{in}}$ | $0. \pm 0$ | $3.8 \pm 1.7$ | $-41.1 \pm 6.3$ | $-0.5 \pm 8.3$ | $-5.0 \pm 3.6$ | $\mathbf{17.4 \pm 1.7}$ | $-0.6 \pm 5.$ | $16.3 \pm 1.1$ |
| | $S^{\text{out}}$ | $0. \pm 0$ | $6.8 \pm 1.8$ | $2.9 \pm 1.6$ | $2.6 \pm 5.9$ | $\mathbf{17.5 \pm 0.7}$ | $16.8 \pm 0.6$ | $17.1 \pm 1.1$ | $17.1 \pm 1.2$ |
| | $S^{\text{pl}}$ | $0. \pm 0$ | $-13.9 \pm 2.2$ | $-6.5 \pm 0.6$ | $\mathbf{0. \pm 0}$ | $\mathbf{0. \pm 0}$ | $-10.8 \pm 4.$ | $\mathbf{0. \pm 0}$ | $\mathbf{0. \pm 0}$ |
| | $S^{-}$ | $0. \pm 0$ | $6.7 \pm 2.6$ | $2.4 \pm 0.9$ | $4.1 \pm 4.9$ | $\mathbf{20.7 \pm 2.9}$ | $18.4 \pm 1.5$ | $\mathbf{20.7 \pm 2.9}$ | $\mathbf{20.7 \pm 2.9}$ |
| | $S^{+}$ | $0. \pm 0$ | $-14.6 \pm 1.4$ | $-7.5 \pm 1.$ | $0. \pm 0$ | $0. \pm 0$ | $-3.0 \pm 1.7$ | $0. \pm 0$ | $\mathbf{0. \pm 0}$ |
| | **Avg.** | $0. \pm 0$ | $-2.2 \pm 11.0$ | $-10. \pm 18.1$ | $1.2 \pm 2.0$ | $6.6 \pm 11.6$ | $7.8 \pm 13.7$ | $7.4 \pm 10.6$ | $\mathbf{10.8 \pm 10.0}$ |

Table 6: Results on CTrL sequences which assess perceptual transfer ($S^{-}$, $S^{\text{out}}$), plasticity ($S^{\text{pl}}$), backward transfer ($S^{+}$) and latent transfer $S^{\text{in}}$.

| | | SA | MNTDP-D | PT-only | NT-only | PICLE |
|---|---|---|---|---|---|---|
| $\mathcal{A}$ | $S^{\text{in}}$ | 58.77 | 61.36 | 61.78 | **63.41** | 63.10 |
| | $S^{\text{out}}$ | 74.25 | 77.95 | **78.15** | - | **78.15** |
| | $S^{\text{pl}}$ | 58.25 | 93.72 | **93.79** | - | **93.79** |
| | $S^{-}$ | 56.28 | 81.67 | **81.92** | - | **81.92** |
| | $S^{+}$ | 73.61 | **74.54** | 74.49 | - | 74.49 |
| | **Avg.** | 64.23 | 77.85 | 78.03 | - | **78.29** |
| $Tr^{-1}$ | $S^{\text{in}}$ | 0. | 22.12 | 24.67 | **32.57** | **32.57** |
| | $S^{\text{out}}$ | 0. | **15.41** | 15.41 | - | 15.41 |
| | $S^{\text{pl}}$ | 0. | **00.20** | 00.20 | - | 00.20 |
| | $S^{-}$ | 0. | **34.29** | 34.29 | - | 34.29 |
| | $S^{+}$ | 0. | 0. | 0. | - | 0. |
| | **Avg.** | 0. | 14.40 | 14.91 | - | **16.49** |

Table 7: The evaluations on the short CTrL sequences using the original experimental setup proposed by Veniat et al. (2020).

presenting the CTrL benchmark states that 7 modules are used, we used the authors' code (Veniat, 2021) for this method which specifies only 6 modules with the same total number of parameters. The difference from the architecture stated in the paper is that the output layer is placed in the last module, instead of in a separate module.

All parameters are trained to reduce the cross-entropy loss with an Adam optimiser Kingma & Ba (2014) with $\beta_1 = 0.9$, $\beta_1 = 0.999$ and $\epsilon = 10^{-8}$. For each task, each path is evaluated 6 times with different combinations of values for the hyperparameters of the learning rate ($\{10^{-2}, 10^{-3}\}$) and of the weight decay strength $\{0, 10^{-5}, 10^{-4}\}$. The hyperameters which lead to the best validation performance are selected. Early stopping is employed during training. If no improvement is achieved in 300 training iterations, the parameters with the the best logged validation performance are selected. Data augmentation is also used during training, namely random crops (4 pixels padding and 32x32 crops) and random horizontal reflection.

The results, presented in Table 7, show that PICLE is able to outperform MNTDP-D on these sequences as well.

