# OpenReview forum: "A Probabilistic Framework for Modular Continual Learning"
_ICLR.cc/2024/Conference — ICLR 2024 poster_

### Official Review · Reviewer_REkX · 2023-10-22

**Soundness:** 4 excellent
**Presentation:** 3 good
**Contribution:** 3 good
**Rating:** 8
**Confidence:** 4

**Summary:**

The authors propose PICLE a probabilistic approach to spawn, compose and train a series of neural modules for solving continual and transfer learning tasks with compositionality. The proposed method considers two separate scenarios:  perceptual transfer, where the input space is the same for all tasks and the output function has to be adapted; and latent transfer, where the output functions have to be reused on new input spaces. For the first scenario, the authors approximate the input distribution of each module with the combination of a low-rank gaussian distribution and the accuracy as the prior. The final path is found by greedy search. For the second scenario, the authors use Bayesian optimization and a Gaussian process to approximate the accuracy of unseen module combinations. One of the main advantages of the proposed approach is that it scales with respect to the number of tasks sinc it evaluates a constant number of compositions and trains a network with a fixed size for each task. The proposed approach achieves state of the art perfomance on CTrL as well as on a new compositional version of CTrL named BELL by the authors.

**Strengths:**

Originality
=======
* The probabilistic modular framework proposed in this work is novel to the best of my knowledge.
* The authors introduce BELL, a new compositional version of CTrL along with this work.

Quality
=====
* The proposed approach is sound.
* The low-rank approximation of the input as a proxy for the likelihood is simple yet effective.
* The appendix contains details to ensure reproducibility and the authors promise to release the code.
* The authors provide Algorithms.
* The authors discuss the limitations of their approach.
* Ablations can be found in the Appendix.

Clarity
=====
* The text is well-written.
* I found Figure 3 and Table 4 very useful in order to understand the difference between PICLE and other algorithms.
* The authors provide details about hyperparameters in the Appendix.

Significance
=========
* Continual learning is a challenging problem and I believe this work is an interesting step towards a modular solution
* PICLE achieves a slight improvement on CTrL and greater improvement on some of the BELL tasks.
* BELL complements CTrL with few-shot and compositional tasks.

**Weaknesses:**

Originality
=======
* LMC also introduced a benchmark for compositional generalization based on colored-mnist that is not mentioned when introducing BELL.
* Moreover the authors claim not to be able to obtain acceptable results with LMC on BELL despite their best efforts, while this is possible, they could have tried to run PICLE on compositional color MNIST task introduced in LMC.

Clarity
=====
* This work introduces a method and a benchmark, however most information about the benchmark is left in the appendix. I suggest providing simpler versions of the Algorithms in the main text, and to use the space to make the paper more self-contained (less dependent on the Appendix).

Significance
=========
* There exists a vast number of continual learning benchmarks. Although they enrich the field, they also dilute the efforts of the research community. Thus I suggest the authors to include some more motivation on why BELL is needed and why researchers should use it rather than other benchmarks or tasks.

Minor
====
* Page 4: Accordignly
* Appendix I: pre-traiend

**Questions:**

* Would it be possible to run PICLE on compositional color MNIST to be able to compare with LMC (see LMC paper)? (It is ok if you do not have enough time / compute resources to do it).
* Could you include some more motivation on why BELL is needed and why researchers should use it rather than other benchmarks or tasks.
* Why $\{\pi}^*$ is not used in Algorithm 2?

---

> ### Author Response · Authors · 2023-11-16
>
> We thank the reviewer for their detailed feedback.
>
> **q2: Could you include some more motivation on why BELL is needed and why researchers should use it rather than other benchmarks or tasks.**
>
> Other CL benchmarks, such as split-CIFAR and CLEAR[2], can differ by the continual learning setup they assume (e.g. class-incremental, domain-incremental etc.). As a result, their sequences implicitly simultaneously evaluate a subset of CL properties, e.g. plasticity, stability and perceptual transfer, which are all summarized by a single number --- the final average accuracy. This makes it difficult to determine which of these properties is lacking from a CL algorithm.
>
> It is important that the benchmarks which we use be able to measure whether algorithms have specific CL properties, like those we list in Table 4. This would guide future research in the field by making the desiderata explicit and facilitating the comparison of different CL algorithms against it.
>
> CTrL [1] is a step in this direction. However, it evaluates perceptual transfer between two problems by shuffling the original problem’s labels, and it evaluates latent transfer between MNIST images with two different background colors. As such, CTrL cannot evaluate perceptual and latent transfer between disparate tasks and input domains. \
> BELL is a further step in that direction, which addresses CTrL’s shortcomings by creating sequences of compositional problems. The problems’ compositionality allows us to create sequences which evaluate perceptual and latent transfer across disparate tasks and input domains. It also allows us to create more challenging sequences for evaluating perceptual transfer --- S^out* and S^out**. BELL also introduces new sequences which evaluate new CL properties --- S^few which evaluates few-shot transfer and S^sp which evaluates latent transfer across different input spaces.
>
> Overall, BELL should be used by researchers in order to diagnose the CL properties of a CL algorithm, which would improve our ability to compare different approaches.
>
> **wC, wS: Making the paper more self-contained, incorporating additional motivation for using BELL**
>
> Thank you for the suggestion, we will add more information and motivation about the benchmark suite to the main text.
>
> **wO1: LMC also introduced a benchmark for compositional generalization based on colored-mnist that is not mentioned when introducing BELL.**
>
> Thank you for pointing this out, we will change the text to mention the LMC sequence for compositional generalization.
>
> LMC introduced a sequence of tasks for evaluating compositional generalization. In it, each problem is derived from MNIST and is defined by one background-foreground color selection and one MNIST label pair. This leads to a limited variety between problems -- input domains are always somewhat similar as they all use MNIST digits, and tasks are restricted to be distinguishing between MNIST classes.
>
> This sequence appears to be the most similar to BELL’s S^few which evaluates a model’s ability to achieve few-shot transfer by recomposing previous knowledge, with S^few involving disparate input domains and output tasks.
>
> **wO2, q1: Running PICLE on LMC's compositional color MNIST.**
>
> We agree that these results would be nice to have. It's unlikely that we will have time to get these results by the end of the discussion period, but we will try and report back if we manage.
>
> **q3: Why is $\mathbf{\pi}^\*$ not used in Algorithm 2**
>
> In Algorithm 2, $\mathbf{\pi}^\*$ is the given list of previous solutions. Our search through PT paths does not need it, which is reflected in it being unused in Algorithm 2. We currently have $\mathbf{\pi}^\*$ as an input in order to keep the inputs to Algorithms 2 and 3 the same. However, we now appreciate that this can cause confusion and will remove $\mathbf{\pi}^\*$ from the inputs of Algorithm 2.
>
> References:\
> [1] Veniat, T., Denoyer, L. and Ranzato, M.A., 2020. Efficient continual learning with modular networks and task-driven priors. arXiv preprint arXiv:2012.12631.\
> [2] Lin, Z., Shi, J., Pathak, D. and Ramanan, D., 2021, August. The clear benchmark: Continual learning on real-world imagery. In Thirty-fifth conference on neural information processing systems datasets and benchmarks track (round 2).

---

> ### Comment · Reviewer_REkX · 2023-11-19
> **Response to rebuttal**
>
> Thank you for your responses. After reading them and those to other reviewers I have decided to raise my score to 8.

---

### Official Review · Reviewer_K341 · 2023-10-30

**Soundness:** 3 good
**Presentation:** 3 good
**Contribution:** 3 good
**Rating:** 8
**Confidence:** 3

**Summary:**

The work proposes a probabilistic modeling approach to efficiently search for the best-fit module composition out of the large discrete space of possible compositions of the modules in continual learning. Depending upon the similarity of the input distributions with that of a previous problem, two variants of the probabilistic model have been proposed: one for perceptual transfer where the prior uses the original accuracy of pre-trained modules to order these, and the other for latent transfer where the prior specifies that pre-trained modules in a path have been used together to solve a previous problem.

**Strengths:**

- Good motivation, presentation, and writing. The equations have been explained well.
- The idea of using the validation accuracy for a path as the proxy for its fitness is simple and elegant.
- The limitations of the proposed method have been elaborated well.
- The reported evaluation metrics are rigorous.

**Weaknesses:**

Please see the questions.

**Questions:**

- On page 3, the authors mention their strategy is based on a generative model of the input x. How is the generative quality of the proposed method quantitatively? Some further evaluation of the proposed method using metrics like ECE can thus be more insightful.

- While I am not very familiar with the up-to-date modular continual learning literature, the baselines in Tables 1-2 look classic to me. Can the authors comment on comparing with more recent works?

- Can the authors compare the computational overhead of their method against the baselines?

---

> ### Author Response · Authors · 2023-11-15
>
> We thank the reviewer for their feedback.
>
> **q1: Quantitative evaluation of the generative quality of the proposed method.**
>
> For our purposes, the employed generative model needs to be useful for distinguishing between inputs sampled from different distributions. As a result, we have not explored the generative quality of the samples. Given our use of efficient Gaussian approximations, we expect that it is poor. Instead, in Appendix I, we have provided a quantitative evaluation of our method’s ability to distinguish between different input domains. The results indicate that using random projections improves the discriminative capabilities of our generative model.
>
> **q2: Baselines in Tables 1-2  appear classic. Can the authors comment on comparing with more recent works?**
>
> Our work aims to improve modular CL methods. Therefore, we compare to the latest state-of-the-art modular CL algorithms, namely: MNTDP-D [1] (2020) and LMC [2] (2021). We also compare to an additional modular CL algorithm -- HOUDINI [3].
> For completeness, as done in previous modular CL work [1, 2], we use O-EWC [4, 5] as a classic regularization-based CL algorithm, and ER [6] as a replay-based method, which was shown to have competitive performance despite its simplicity.
>
> Tables 1-2 do not report the performance of LMC since it performed poorly, despite our efforts to adjust it to the BELL benchmark suite.
>
> **q3: Can the authors compare the computational overhead of their method against the baselines?**
>
> Figure 3a of our submission compares the computation demand of PICLE to those of the baselines, evaluated in FLOPs. It can be seen that since PICLE uses slightly more resources than MNTDP-D since it considers a larger search space. However, it can also be observed that PICLE’s computational requirements scale well with the number of solve problems.
>
> References: \
> [1] Veniat, T., Denoyer, L. and Ranzato, M.A., 2020. Efficient continual learning with modular networks and task-driven priors. arXiv preprint arXiv:2012.12631. \
> [2] Ostapenko, O., Rodriguez, P., Caccia, M. and Charlin, L., 2021. Continual learning via local module composition. Advances in Neural Information Processing Systems, 34, pp.30298-30312. \
> [3] Valkov, L., Chaudhari, D., Srivastava, A., Sutton, C. and Chaudhuri, S., 2018. Houdini: Lifelong learning as program synthesis. Advances in neural information processing systems, 31. \
> [4] Schwarz, J., Czarnecki, W., Luketina, J., Grabska-Barwinska, A., Teh, Y.W., Pascanu, R. and Hadsell, R., 2018, July. Progress & compress: A scalable framework for continual learning. In International conference on machine learning (pp. 4528-4537). PMLR. \
> [5] Chaudhry, A., Dokania, P.K., Ajanthan, T. and Torr, P.H., 2018. Riemannian walk for incremental learning: Understanding forgetting and intransigence. In Proceedings of the European conference on computer vision (ECCV) (pp. 532-547). \
> [6] Chaudhry, A., Rohrbach, M., Elhoseiny, M., Ajanthan, T., Dokania, P.K., Torr, P.H. and Ranzato, M.A., 2019. On tiny episodic memories in continual learning. arXiv preprint arXiv:1902.10486.

---

> > ### Comment · Reviewer_K341 · 2023-11-19
> >
> > The authors have addressed the majority of my concerns. I am thus raising my score. Good work. I have no further questions.

---

### Official Review · Reviewer_g7xb · 2023-11-01

**Soundness:** 3 good
**Presentation:** 3 good
**Contribution:** 3 good
**Rating:** 8
**Confidence:** 4

**Summary:**

This paper introduces PICLE, a scalable modular continual learning algorithm that addresses key challenges in the field. PICLE is task-aware and optimizes for different types of transfer learning, such as perceptual, few-shot, and latent transfer. Utilizing a probabilistic search method, it efficiently approximates the fitness of different module compositions, significantly reducing training requirements. The algorithm outperforms existing state-of-the-art solutions, demonstrating its efficacy on the popular CTrL benchmark suite and a new extension called BELL.

**Strengths:**

1. Modularity in continual learning is highly promising, offering a good balance between scalability and the ability to transfer knowledge across tasks without undue interference.

2. The key challenge in modular approaches is the exponentially increasing search space for module combinations, particularly as the number of layers grows. PICLE addresses this scalability issue effectively, allowing for both perceptual and latent transfer, which makes it a standout in the field.

3. The paper's empirical study is both comprehensive and well-executed, adding robustness to its claims.

**Weaknesses:**

Minor weaknesses, if addressed during the rebuttal, I will keep my score at an 8:

1. The table in the paper lacks uncertainty metrics such as standard deviation. This omission should be addressed to enhance the study's reliability. Additionally, for readability purposes, it would be better to show the percentages only up to three digits instead of four (e.g., XX.XX% should be changed to XX.X%).

2. The paper should clearly state that the PICLE method is task-aware, which is an important limitation. Ideally, there would be a column in Table 4 that discusses task-agnosticism, a feature that another algorithm, LMC, is capable of.

3. The concept of using a generative model to approximate latent activations and assume local independence from one layer to the next, as introduced in Section 4, was initially proposed in the LMC paper. The authors should give credit to this paper and clarify how their methods differ from those originally proposed in the LMC paper.

**Questions:**

None

---

> ### Author Response · Authors · 2023-11-15
>
> We thank the reviewer for their feedback.
>
> **w1a: Lack of uncertainty metrics in the table.**
>
> As the performance of the different algorithms on the short sequences of the CTrL benchmark is similar, we added the standard deviations of the results shown in Table 3. This can be found in Table 6, Appendix K.1. Does this address your concern?
>
> **w1b: Show percentages for up to 1 decimal point.**
>
> We agree that this would improve the paper’s readability and will make the changes.
>
> **w2: The paper should clearly state that the PICLE method is task-aware**
>
> We agree that this is an important property, which provides an interesting distinction between the algorithms. Therefore, we will mention it and add it as a row in Table 4.
>
>
> **w3: Credit to and better distinction from LMC.**
>
> We have credited LMC with a similar approach for perceptual and few-shot transfer in the Related Work section, while also outlining the differences. However, we agree with the wording proposed by the reviewer, and will also add it to Section 4 for better visibility. Moreover, we acknowledge that the conceptual differences can be better presented and will amend the text accordingly.
>
> The main conceptual difference is that while LMC also uses a module-specific generative model, PICLE unites the generative model for each layer using a single probabilistic model, which brings numerous advantages. First, our probabilistic model allows us to define a prior over the choice of modules. Second, by changing the probabilistic model, we can incorporate further assumptions in a principled way, for example, we can put a prior over the choice of pre-trained modules for multiple layers, rather than layer-specific priors.
> Third, the probabilistic model allows us to compute the posterior over the choice of pre-trained module for each layer collectively, rather than individually.
>
> The main implementation difference is that PICLE utilizes a different approximation of a module’s input distribution, using orders of magnitude fewer extra parameters per pre-trained module than LMC.

---

### Official Review · Reviewer_gkN4 · 2023-11-02

**Soundness:** 3 good
**Presentation:** 3 good
**Contribution:** 3 good
**Rating:** 6
**Confidence:** 4

**Summary:**

This paper addresses the modular continual learning setting.  They introduce a probabilistic model to determine a distribution over paths for perceptual transfer.  This can be interpreted as selecting modules based on how similar the current features at that layer are to the modules' input distribution.  For latent transfer, they have a probabalistic model based on the idea that suffixes similar in L2 distance should have similar performance.  This modeling allows predicting the validation performance of a path without training.  Due to the modeling, they only need to evaluate a number of paths which grows linearly with the depth L.  Their method outperforms the other methods marginally on average, while in some cases demonstrating significant boosts, e.g. few-shot transfer.

**Strengths:**

* The probabalistic models introduced are simple and make sense.  Using such modeling to avoid the expensive counterfactual of evaluating a path is a logical approach.
* The paper is well written and I found the appendix helpful.
* The evaluations seem consistent with prior works

**Weaknesses:**

* The quantitative advantage over MNTDP-D in accuracy is marginal on average ~1-3%
* The method requires slightly more FLOPs than MNTDP-D (Figure 3 (a))

Overall I am voting to accept this paper due to the conceptual contribution, however the quantitative results seem marginal to me, and thus I am not voting for a stronger accept.

**Questions:**

**Table 1:** Why are the numbers for MNTDP-D and PICL forward transfer so similar?

Typo: “The CL algorithm should be able to ”remember” previous problems“ (backwards quotation)

**Algorithm 3:** Bold lambda on right side of line 5

“This search results in the most relevant previous solution $\pi'$. Finally, in lines 11-14, we evaluate NT paths created by transferring a different number of the last $\ell \in \\{\ell_{min} + 1, ..., L − 1\\}$ layers of $\pi'$, to see if re-using more layers leads to further improvement”. (do you mean $\ell \in \\{2, \ldots L - 1\\}$?  That's what the code appears to be searching over

---

> ### Author Response · Authors · 2023-11-15
>
> We thank the reviewer for their feedback. We will correct the reported typos.
>
> ## Comparing MNTDP-D and PICLE:
>
> We would argue that this is a case where looking at the average accuracy across tasks is misleading. Table 4 and Appendix H of our submission list a set of desirable properties for CL methods. Each sequence from our experiments evaluates a different CL property. Our argument is that MNTDP-D and PICLE share many desirable properties (see Table 4), so they will have similar accuracy on sequences that test those properties. But PICLE has a few desirable properties that MNTDP-D does not.
>
> As a result, MNTDP-D and PICLE are expected to and do perform similarly on the sequences which evaluate: perceptual transfer (S^-, S^out, S^out*), plasticity (S^pl), Stability (S^+). We note that, PICLE is able to demonstrate perceptual transfer on S^out**, Table 1 while MNTDP-D fails to do so, leading to PICLE achieving **+10.3** higher transfer on the last problem.
>
> However, PICLE is expected to and does significantly outperform MNTDP-D on few-shot  transfer and latent transfer. Since the short sequences were designed to evaluate different transfer learning properties by an algorithm’s performance on the last problem, we consider the amount of transfer to the last problem. For few-shot transfer, PICLE achieves **+34.65** higher transfer (S^few, Table 1) than MNTDP. For latent transfer, PICLE also demonstrates much higher transfer: **+12.58** (S^in, Table 2), **+23.65** (S^sp, Table 2), **+16.89** (S^in, Table 3).
>
> We discuss this in Section 6, however, thanks to your comments, we will change the text to state this more clearly.
>
> **w1: The quantitative advantage over MNTDP-D in accuracy is marginal on average ~1-3% \
> q1: “Why are the numbers for MNTDP-D and PICL forward transfer so similar?”**
>
> Since PICLE and MNTDP-D share many CL properties, apart from few-shot transfer and latent transfer, their performance --- including accuracy and forward transfer --- on the short sequences is similar, apart from on S^few, S^in, S^sp.
>
> In Table 1, the accuracy is averaged over mostly shared CL properties (apart from few-shot transfer). PICLE’s advantage is only apparent by its superior performance on the last problem of S^few and S^out**. As a result, the average accuracy in Table 1 averages over 42 problems, while PICLE achieves significant improvement over MNTDP-D on 2 of them. This leads to the similar average accuracy which we observe. Similarly, the short sequences of CTrL mostly evaluate CL properties which are shared between MNTDP-D and PICLE, leading to PICLE outperforming MNTDP-D only on the last problem of S^in, and in turn leading similar average accuracies in Table 3.
>
> PICLE’s advantage over MNTDP-D is that it can achieve few-shot and latent transfer. Overall, PICLE achieves much higher transfer on the last problem than MNTDP-D on: perceptual transfer (S^out**, Table 1, **+10.3**), few-shot transfer (S^few, Table 1, **+34.65**), latent transfer (S^in, Table 2, **+12.58**), (S^sp, Table2, **+23,65**), (S^in, Table 3, **+16,89**).
>
> **w2: The method requires slightly more FLOPs than MNTDP-D (Figure 3 (a))**
>
> This is correct. Figure 3(a) shows that PICLE requires slightly more FLOPs, while considering a much bigger bigger search space. The figure also shows that the computational demands of both PICLE and MNTDP-D scale well with the number of problems.
>
> ## Other questions:
> **q2: “Algorithm 3 - Do you mean $\ell \in \{ 2, ..., L-1 \} $?”**
>
> The $\ell$ defined on line 11 of Algorithm 3 refers to the index of the first pre-trained module. On line 12 we use it to extract the pre-trained modules $\pi’[\ell : L]$. For $\ell = \ell_{min}$ , the length of the pre-trained suffix $\pi’[\ell_{min} : L]$ is $\ell_{min} + 1$. And for $\ell = 2$, the length of the pre-trained suffix $\pi’[2 : L]$ is $L-1$. Therefore, we indeed attempt to transfer ${\ell_{min}+1, ..., L-1}$ pre-trained layers.
> However, thanks to your comment, we now see that it is confusing to use $\ell$ inside the text to denote length while simultaneously using it in Algorithm 3 to denote an index. We will adjust the text accordingly.

---

> > ### Comment · Reviewer_gkN4 · 2023-11-21
> > **Response to authors part 1**
> >
> > I thank the authors for their response and clarification.  I maintain my recommendation to accept this work.

---

### Meta-Review · Area_Chair_dPpC · 2023-12-11

**Metareview:**

(a) The paper proposes a probabilistic modular continual learning framework, dubbed as PICLE. It enables to reuse some pre-trained modules from previous task to increase efficiency in continual learning.
(b) Conceptually the authors proposed new framework for modular continual learning (combining perceptual transfer / few-shot transfer / scalable latent transfer) via casting it to a best path finding problem.
(c) The weakness is that the performance gain over previous work, e,g, MNTDP-D, seems marginal for some cases.

**Justification For Why Not Higher Score:**

Despite the conceptual contribution, the overall performance gain over the previous work does not seem significant, but rather marginal some times. It would be good for the authors to highlight the clear gain of PICLE over MNTDP-D in the final version as they have described in the rebuttal.

**Justification For Why Not Lower Score:**

The method is novel and worth publishing at ICLR conference as all the reviewers have mentioned.

---

### Decision · Program_Chairs · 2024-01-16

Accept (poster)